# Zebrafish optic nerve injury results in systemic retinal ganglion cell dedifferentiation

**Ashrifa Ali**[1], **Hannah Schriever**[2], **Dennis Kostka**[2], **Takaaki Kuwajima**[3],
**Kristen M. Koenig**[1]*, **Jeffrey M. Gross**[1]*

**1** Department of Molecular Biosciences, The University of Texas at Austin, Austin, Texas, United States of America, **2** Department of Computational and Systems Biology, The University of Pittsburgh Medical School, Pittsburgh, Pennsylvania, United States of America, **3** Department of Ophthalmology, Louis J. Fox Center for Vision Restoration, The University of Pittsburgh Medical School, Pittsburgh, Pennsylvania, United States of America

* jmgross@austin.utexas.edu (JMG); kmkoenig@utexas.edu (KMK)

## Abstract

Retinal ganglion cells (RGCs) are the sole projection neurons connecting the retina to the brain and therefore play a critical role in vision. Death of RGCs during glaucoma, optic neuropathies and after ocular trauma results in irreversible loss of vision as RGCs do not regenerate in the human eye. Moreover, there are no FDA approved therapies that prevent RGC death and/or promote RGC survival in the diseased or injured eye. There is a critical need to better understand the molecular underpinnings of neuroprotection to develop effective therapeutic approaches to preserve damaged RGCs. Unlike in mammals, RGCs in zebrafish are resilient to optic nerve injury, even after complete transection of the optic nerve. Here, we leveraged this unique model and utilized single-cell RNA sequencing to characterize RGC responses to injury and identify putative neuroprotective and regenerative pathways. RGCs are heterogeneous and studies in mice have shown that there is differential resiliency across RGC subtypes. Our results demonstrated that all RGC subtypes are resilient to injury in zebrafish. Quantifying changes in gene expression revealed the upregulation of progenitor and regenerative markers in all RGC subtypes after injury as well as distinct early and late phases to the injury response. This shift in gene expression causes injury-responsive RGCs to resemble RGC subtype 3, a low frequency population of endogenous immature RGCs that are normally maintained in the wild-type, uninjured adult retina. A similar but restricted transcriptomic injury response in RGCs of the uninjured contralateral eye was also detected, highlighting a systemic RGC response to unilateral optic nerve injury. Taken together, these results demonstrate that zebrafish RGCs dedifferentiate in response to injury, and this may be a novel mechanism mediating their unique cell survival and regenerative capabilities.

**Data availability statement:** All sequencing data have been deposited to the NCBI Gene Expression Omnibus (GEO) under the following accession numbers: RNA-Seq:GSE284728 scRNA-Seq:GSE284729.

**Funding:** This work was supported by a Knights Templar Eye Foundation Pediatric Ophthalmology Career-Starter Research Grant (A.A.) (https://www.ktef.org/grants) and the University of Texas at Austin (J.M.G.). The funders played no role in study design, data collection and analysis, decision to publish or preparation of the manuscript.

**Competing interests:** The authors have declared that no competing interests exist.

## Author summary

Retinal ganglion cells (RGCs) connect the eye to the brain and are essential for vision. Their death in conditions like glaucoma, affecting over 70 million people worldwide, leads to permanent blindness, with no FDA-approved treatments to prevent it. Unlike mammals, zebrafish RGCs are resilient to optic nerve injury. In this study, we used next-generation sequencing technologies to characterize the RGC response to optic nerve injury at the single-cell level. We discovered that all zebrafish RGCs survive damage by temporarily shifting into a less mature state, resembling a rare population of immature RGCs found in uninjured animals. We identified many genes whose expression changes early or late in the injury response as well as a similar but restricted transcriptomic injury response in the uninjured contralateral RGCs, highlighting the systemic RGC response to optic nerve injury. This work is significant because our detailed characterization of RGC responses to optic nerve injury identifies dedifferentiation as an injury response, possibly important for cell survival and axon regrowth. The genes and pathways we identify are potential therapeutic targets to enable RGC survival in the injured or diseased human eye.

## Introduction

Visual information is processed in the retina and relayed to the brain via the optic nerve, which is composed of bundled retinal ganglion cell (RGC) axons. The optic nerve is highly vulnerable to damage and is compromised in neurodegenerative diseases like glaucoma, as a result of ocular trauma or in optic neuropathies. While substantial progress has been made in identifying the molecular and cellular events that lead to RGC death and in identifying potential neuroprotective strategies that could maintain the health of RGCs after axonal damage, no FDA-approved therapies have been identified that promote RGC survival after optic nerve injury [1]. Likewise, while many molecules that enhance RGC axon regeneration post-injury have been identified in a variety of experimental systems, none have yet been shown to be clinically effective to regenerate axons in the human eye [2–5]. There remains a critical need to better understand the molecular underpinnings of RGC neuroprotection and axon regeneration such that these can be leveraged to develop therapeutic approaches that preserve or restore RGCs lost to neurodegenerative disease and ocular injury.

Zebrafish have emerged as an excellent model system to study ocular diseases [6,7]. Moreover, they have a remarkable intrinsic ability to regenerate retinal neurons that have been damaged [8,9], as well as to both protect RGCs from death after optic nerve damage, and to regenerate RGC axons when the optic nerve is injured [10]. Work by many laboratories, including our own, has begun to identify the molecular underpinnings of these intrinsic neuroprotective and regenerative responses (e.g., [11–20]). Previously, our lab reported that zebrafish RGCs survive severe optic nerve injury, and that Jak/Stat pathway activity mediates RGC survival [14]. Despite these

advances, we still know little about the mechanisms mediating RGC survival or how RGCs transition towards axon regeneration post-injury.

RGC subtypes have been characterized in several organisms [21–24] including zebrafish [25]. While nearly all RGCs die in mice after optic nerve crush, there is a small percentage that do survive and subtype specific resiliency has been identified [26,27]. Further studies of resilient subtypes revealed changes in gene expression that are thought to confer neuroprotective and pro-regenerative abilities upon them [26]. Given that most zebrafish RGCs survive optic nerve transection, we sought to leverage this system and utilize single cell RNA sequencing (scRNA-Seq) to determine if there is any heterogeneity in their intrinsic neuroprotective abilities and to further identify genes and pathways contributing to RGC survival.

## Results

### Single cell transcriptional analysis identifies molecular RGC subtype identities

To assess the transcriptional response of RGCs to injury, we performed optic nerve transection (ONT) on the left eye of 3–6 month old *isl2b*:GFP fish [14]. We then dissected both left and right eyes, injured and uninjured respectively, at one- and seven-days post-injury (dpi). We chose 1dpi as gene expression changes occur rapidly in RGCs after optic nerve injury (e.g., [28,29]) while at 7dpi, gene expression changes associated with axon regeneration are robust and axon regeneration is underway [8,30]. GFP positive cells were isolated by FACS and prepared for single-cell RNA sequencing (scRNA-Seq) (Fig 1A). After computational analyses and removal of contaminating cell types (see Methods for additional details), we recovered and profiled a total of 17,769 RGCs. Uniform manifold approximation and projection (UMAP) dimensional reduction and clustering identified 55 *de novo* clusters. A previous study characterized RGC diversity in naive larval and adult zebrafish [25]. Here, we generated both an integrated analysis, including these previously published RGC data in our analysis, and we performed an independent analysis of our data, using a label transfer step to align our data to the published wildtype RGC subtype identities (Fig 1B and 1C). We found that the previously published RGC single cell dataset aligned well to our Uninjured Day 1 dataset (S1 Fig) and; therefore, in order to avoid batch effects, we used Uninjured Day 1 as our control for this study. All previously identified RGC subtypes were detected in our dataset, with the exception of subtype 32, the rarest subtype identified in the previous study. Subtypes 24, 26 and 30 from the previous study were combined into a single cluster in our analyses, and two additional RGC subtypes were identified, which we termed n1 and n2.

The mouse retina progressively loses RGCs in response to optic nerve crush [31]. After the crush injury, RGCs of all subtypes are lost; however, it has been shown that specific RGC subtypes are more resilient to injury where they have higher proportions that survive [26]. All zebrafish RGC subtypes were identified in 1 and 7dpi single cell datasets, suggesting that all RGC subtypes are resilient to ONT in zebrafish (Figs 1C and S1). We next identified differential gene expression across all RGC subtypes, with subtype-enriched genes paralleling those previously reported (Figs 1D and S2 and S1 Table) [25]. While zebrafish RGCs have been characterized physiologically (e.g., [32]), and unique subtypes identified (e.g., a UV-responsive ON-sustained subtype; [33]), the intersection of molecular identity and physiological function is only beginning to be understood [25]. Segregating our samples by experimental condition (Uninjured Day 1; Uninjured Day 7; Injured Day 1 and Injured Day 7) began to reveal variable expression in response to injury. Most notably, all RGC subtypes in Injured Day 7 samples showed high expression of the immature RGC genes *tubb5*, *alcamb* and *tmsb*, generally associated with RGC subtype 3 identity, and showed moderate downregulation of the mature RGC markers *bhlhe40*, *fam107b* and *bhlhe41* (Fig 1E).

### All RGC subtypes survive optic nerve transection in zebrafish but there is an increase in RGC subtype 3 in response to injury

After identifying all RGC subtypes and evidence of injury-dependent changes in gene expression across conditions, we performed differential abundance analysis, quantifying the representation of each RGC subtype within each dataset. We

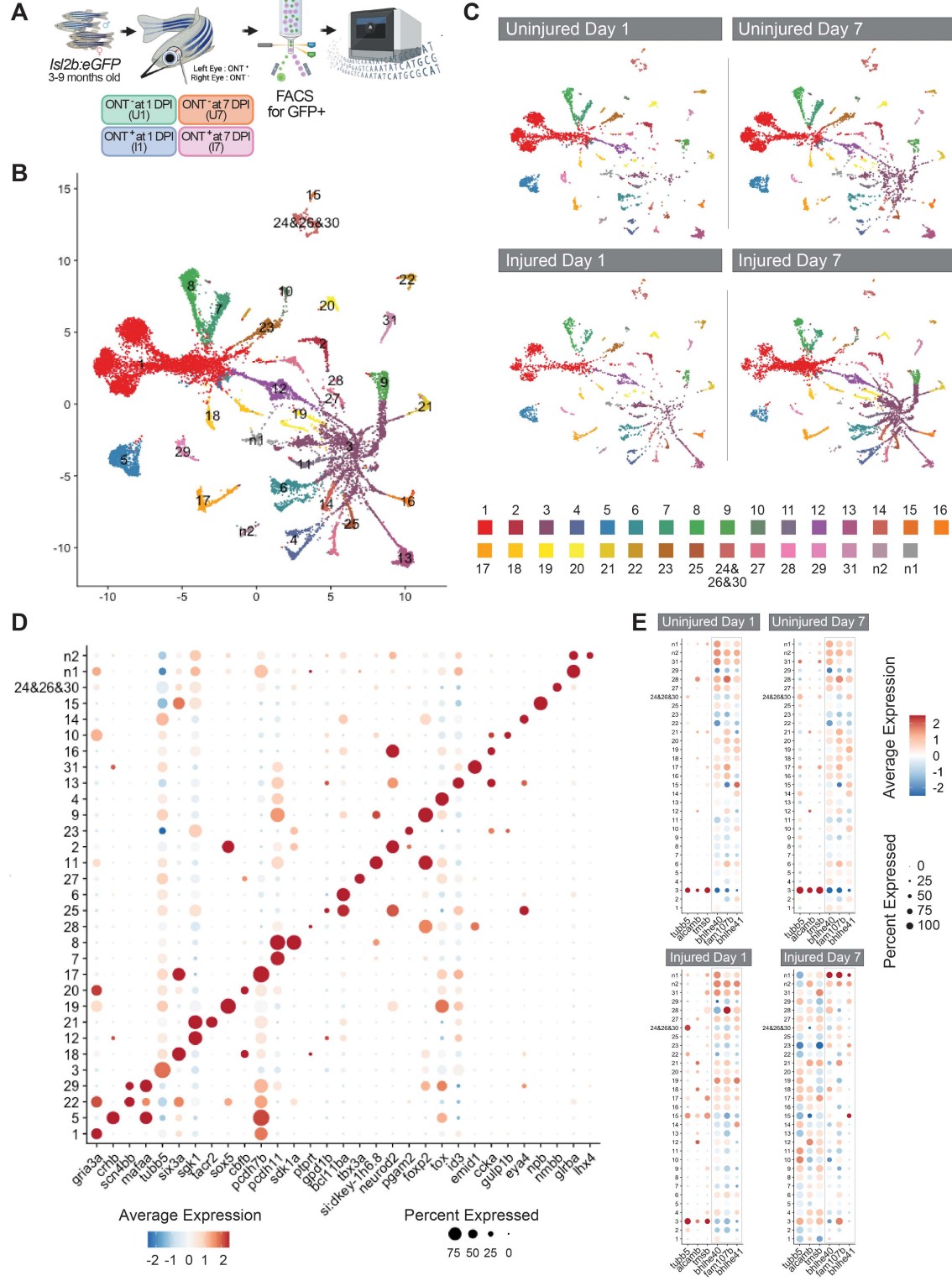

**Fig 1. Single cell transcriptional analysis identifies molecular RGC subtype identities.** A) Experimental design for the single cell sequencing project. Optic nerve transection (ONT) was performed on the left eye of 3-9 month old *isl2b*:GFP transgenic fish. Both left and right eyes were dissected 1 day post injury (DPI) and 7 days post injury. Cells were dissociated and prepared in single cell suspension and sorted for GFP expression. 10x

Genomics single cell sequencing libraries were prepared and sequenced. Created in BioRender. Ali, A. (2025) https://BioRender.com/rgsbfro. B) Uniform manifold approximation and projection (UMAP) of all data generated in this study, clustered and labeled by previously defined retinal ganglion cell subtype labels (1-31) as well as two undescribed subtypes (n1 and n2) [25]. C) UMAP of data generated in this study, split by each dataset: Uninjured Day 1, Uninjured Day 7, Injured Day 1 and Injured Day 7. D) Dot plots of differential gene expression across RGC subtypes. E) Dot plots of gene expression of immature and mature RGC genes segregated by dataset: Uninjured Day 1, Uninjured Day 7, Injured Day 1 and Injured Day 7.

found that all RGC subtypes were represented in all datasets, further supporting a model in which all RGC subtypes are resilient to injury in zebrafish (Figs 2, S1, and S3). However, we found that multiple RGC subtypes showed a decrease in abundance after injury, including subtypes 23, 5, 22, 29, 14, 13, 27, 9, 17 and the combined 24/26/30 subtype cluster. (Fig 2 and S2 Table). Our differential abundance analysis also detected a significant increase in RGC subtype 3 in Uninjured Day 7, Injured Day 1, and Injured Day 7 datasets (Fig 2 and S2 Table). Although losses of these specific RGC subtypes were statistically significant, we were not convinced that this was evidence of biologically relevant subtype susceptibility

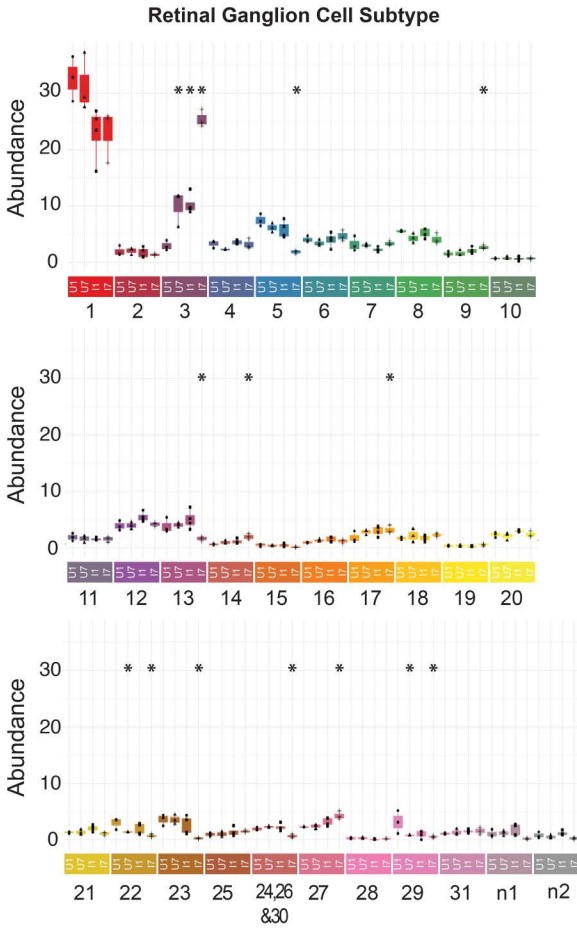

**Fig 2. All zebrafish RGC subtypes survive ONT but subtype 3 increases in response to injury.** Box plot graph of subtype representation within each dataset. The asterisks identify statistically significant changes (P value < 0.02 and F value > 5) as compared to Uninjured Day 1. Each replicate within each experimental condition (U1, U7, I1, I7) is represented by an individual black mark (dot, triangle, box, or plus). All statistical analyses can be found in S2 Table.

to injury similar to that observed in mouse because the actual changes and representation were so small. The strongest signal from our analysis was the significant increase in subtype 3 representation after injury.

The loss and increase in specific RGC subtypes could be the result of selective RGC death and/or RGC trans-differentiation (i.e., the shift in gene expression profile from one RGC subtype to another). Previous work by us [14] and others [10] indicated that while most zebrafish RGCs are resilient to injury, ~25% of RGCs die by 7dpi after optic nerve transection. Our previous studies utilized a decrease in *isl2b*:GFP expression as a proxy for RGC cell survival after injury [14]. Our single-cell data showed a universal downregulation of *isl2b* in RGCs after injury, however (S3 Fig). Thus, it is unclear whether the low-GFP expressing *isl2b*+ RGCs actually die or just transiently downregulate the *isl2b*:GFP transgene. Given our scRNA-Seq results indicating limited changes in RGC subtype abundances post-injury, we wanted to directly assess these contradictory observations and therefore performed TUNEL staining to quantify RGC apoptosis after optic nerve transection. As expected, we did not detect any TUNEL+ cells in the naive retinae. Likewise, we detected very few TUNEL+ cells at either 1 or 7dpi (S4 and S5 Figs). We utilized NMDA injections as a positive control, which triggers rapid apoptosis of RGCs as well as amacrine cells [34,35] to ensure that our TUNEL protocol was effective in whole zebrafish retinae. Numerous TUNEL+ RGCs were detected in NMDA-injected retinae (S4 and S5 Figs). Given these contrary results, we next quantified nuclei in the ganglion cell layer, in naive retinae and at 7dpi in the uninjured and injured eye (S6 Fig). These results showed no changes in the number of RGCs per unit area between the naive and uninjured eye, but a small but significant decrease in the 7dpi injured eye. The ganglion cell layer swells after injury [36] and nucleus size in the 7dpi ganglion cell layer was increased by 39% after injury, however, possibly influencing cell counts (S6 Fig). Taken together, these data indicate that some RGCs are lost after injury, but, most RGCs and importantly, RGCs of all subtypes, survive to 7dpi after ONT, suggesting that RGCs may be transitioning in identity as part of their injury response.

To assess changes in transcriptional profile representation in our data that may be obscured by assessing differential abundance using RGC subtype identity, we next performed abundance analyses using the Milo statistical framework. Milo uses k-nearest neighbor graphs to assess changes in neighborhood occupancy to identify more subtle transcriptional changes across experiments [37]. Milo analyses showed few changes in cluster-assigned neighborhoods when comparing Uninjured Day 1 versus Injured Day 1 datasets (Fig 3A and 3B). However, comparing Uninjured Day 1 and Injured Day 7 datasets showed significant differences in neighborhood occupancy, highlighting an increase in Injured Day 7 cell occupancy in subtype 3 neighborhoods (Fig 3C and 3D), further supporting our previous differential abundance results. Interestingly, when we compared Uninjured Day 1 and Uninjured Day 7 datasets we also detected a significant increase in cells found in subtype 3 assigned neighborhoods in Uninjured Day 7 RGCs. This observation highlighted a systemic response to optic nerve injury found in the contralateral RGCs and that this response also involves the adoption of a subtype 3-associated transcriptional profile upon injury (Fig 3E and 3F).

## RGC subtype 3 possesses a progenitor and regenerative identity

Previous work has suggested that subtype 3 RGCs are post-mitotic, immature RGCs that persist into adulthood [25]. With the significant increase in subtype 3 abundance in response to injury, we were therefore interested in further characterizing the expression profile of this RGC subtype. To begin to assess this, we identified subtype-specific differentially expressed genes. This analysis was performed relative to all other RGC subtypes. Consistent with previous studies, our data showed an enrichment of immature marker gene expression in subtype 3, including the markers: *tubb5, alcamb* and *tmsb* (Figs 4A and S7) [25]. In addition to these genes, we also detected enrichment of RGC regeneration and cell reprogramming-related markers that included: *prph* [38], *sox11a* [39–43], *klf7b* [15,43,44], *mych* [45,46], and *gap43* [47] (Fig 4A). Regeneration and injury response gene enrichment in RGC subtype 3 was further highlighted by KEGG and GO analyses, which showed statistically significant enrichment of processes such as 'axon regeneration', 'response to axon injury' and 'nervous system development' (Fig 4B and 4C). Dotplots of enriched genes associated with 'regenerative

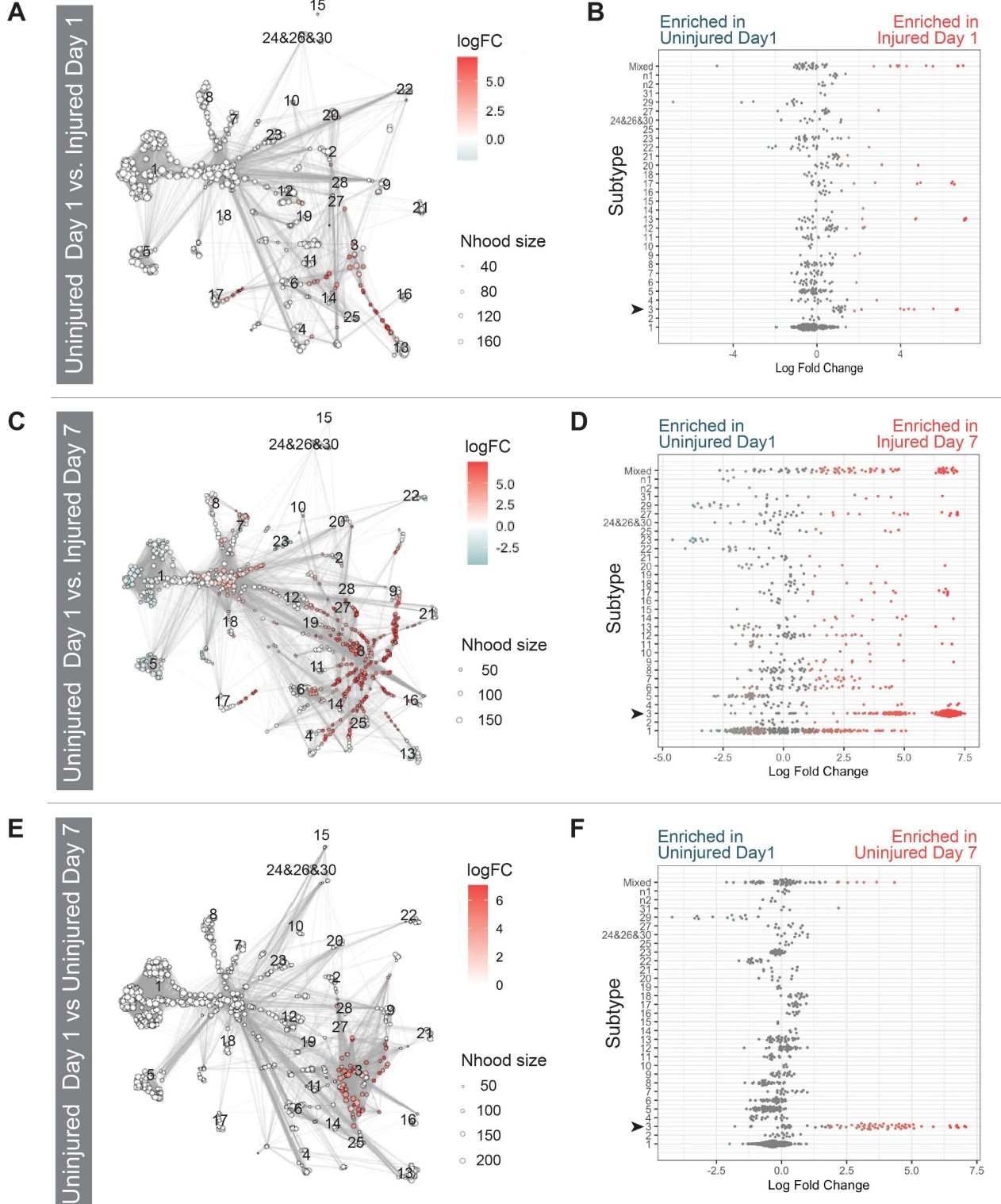

**Fig 3. Subtype 3 RGCs increase in response to injury.** A,C,E) Milo analysis showing differential abundance testing in a k-nearest neighbor graph. Red nodes are neighborhoods that are more abundant in the second dataset, blue nodes are less abundant, and white nodes do not show a significant change. Dot size corresponds to the number of cells in the neighborhood. Graph edges depict the number of cells shared between neighborhoods. B,D,F) Beeswarm plots showing distribution of log fold change in abundance of neighborhoods within RGC subtypes. A,B) Little change is observed in

RGC subtype abundance between Injured Day 1 and Uninjured Day 1. C) Subtype abundance around RGC subtype 3 increased in Injured Day 7 compared to Uninjured Day 1. D) There is a neighborhood enrichment cluster in subtype 3. In addition, there is enrichment shown in mixed neighborhoods, which includes neighborhoods that occupy more than one subtype cluster. E,F) Subtype abundance around RGC subtype 3 increased in Uninjured Day 7 compared to Uninjured Day 1 mirroring the injury response, albeit to a lesser degree.

processes' and 'nervous system development' terms show the specificity of this immature identity to subtype 3 relative to other RGC subtypes (Fig 4D and 4E).

## RGCs dedifferentiate in response to optic nerve injury

Elevated apoptosis does not account for the increased subtype 3 representation in our differential abundance analysis after injury (S4 and S5 Figs). We therefore hypothesized that transcriptionally, RGCs are responding to optic nerve transection by dedifferentiating and becoming more like immature RGCs found in the subtype 3 cluster. To test this hypothesis, we performed a cell potency assessment using CytoTRACE2 (Fig 5A) [48]. CytoTRACE2 is a deep learning framework that evaluates the transcriptome of each cell to assess absolute developmental potential. As expected, subtype 3 RGCs possessed the highest CytoTRACE2 potency score, indicating that they are the least differentiated of the RGC subtypes (Fig 5B). CytoTRACE2 analyses also revealed an increase in potency score in injured RGCs relative to uninjured RGCs, peaking in Injured Day 7 samples, indicating that globally, RGCs in the Injured Day 7 retina are the least differentiated (Fig 5C). If RGCs dedifferentiate in response to optic nerve injury, we wanted to determine the transcriptional trajectory that they take as they dedifferentiate into subtype 3 RGCs. To do this, we performed a pseudotime analysis using Monocle3 rooted in the subtype 3 cluster [49] (Fig 5D and 5E). This analysis specifically highlighted that subtypes 14, 9, 27, 31, 19, 25, and 17 are the closest in pseudotime to subtype 3 identity. These subtypes also have the highest relative CytoTRACE2 scores. Our pseudotime analysis also suggests that subtype 7, 8, and 23 are late in pseudotime. However, our trajectory analysis suggests that these subtypes are terminal in their own branched trajectories and are likely not directly dedifferentiating into subtype 3 RGCs. Collectively, these analyses reveal that some subtypes of RGCs dedifferentiate in response to optic nerve transection and identifies the potential transcriptional path that they adopt to become more like subtype 3.

## Subtype 3 RGCs are intrinsically heterogeneous and contain immature and injury-responsive subpopulations

To better understand RGC dedifferentiation in response to injury, we wanted to more thoroughly analyze subtype 3 cells in our experimental dataset. Subtype 3 RGCs have an immature identity, however, our initial analysis of differentially expressed genes does not investigate the differences between endogenous subtype 3 RGC and the injury response observed. To achieve a greater understanding of this, we first subsetted and re-clustered subtype 3 RGCs (Fig 6A) revealing nine subclusters (Fig 6B and S3 Table). We assessed differential gene expression found across the subclusters. The outcome of that analysis finds genes that are only enriched relative to other subtype 3 RGCs, not the entire dataset as a whole. We found distinct differences in subcluster occupancy based on experimental condition (Fig 6C and 6D). Uninjured Day 1 cells exclusively occupied two endogenous immature RGC populations, subcluster B and subcluster I, both of which showed enrichment of developmentally-expressed genes (Fig 6E). Uninjured Day 7 and Injured Day 1 cells also occupied these subclusters (Fig 6D). This suggested that the endogenous subtype 3 populations are consistent in transcriptional identity and are present in uninjured eyes as well as during the early injury response. Injured Day 7 and Uninjured Day 7 cells populate the remaining 7 subclusters; the prevalence of Uninjured Day 7 cells within these subclusters further highlights the systemic response to injury that occurs in the uninjured contralateral eye. Amongst these seven subclusters, there is a late injured-eye response subcluster, subcluster A, and there is a late uninjured-eye response subcluster, subcluster D, as these are primarily occupied by Injured Day 7 or Uninjured Day 7 cells, respectively. There is also an immune response subcluster, subcluster G, primarily occupied by Injured Day 7 cells (Fig 6E). The remaining four subclusters share marker

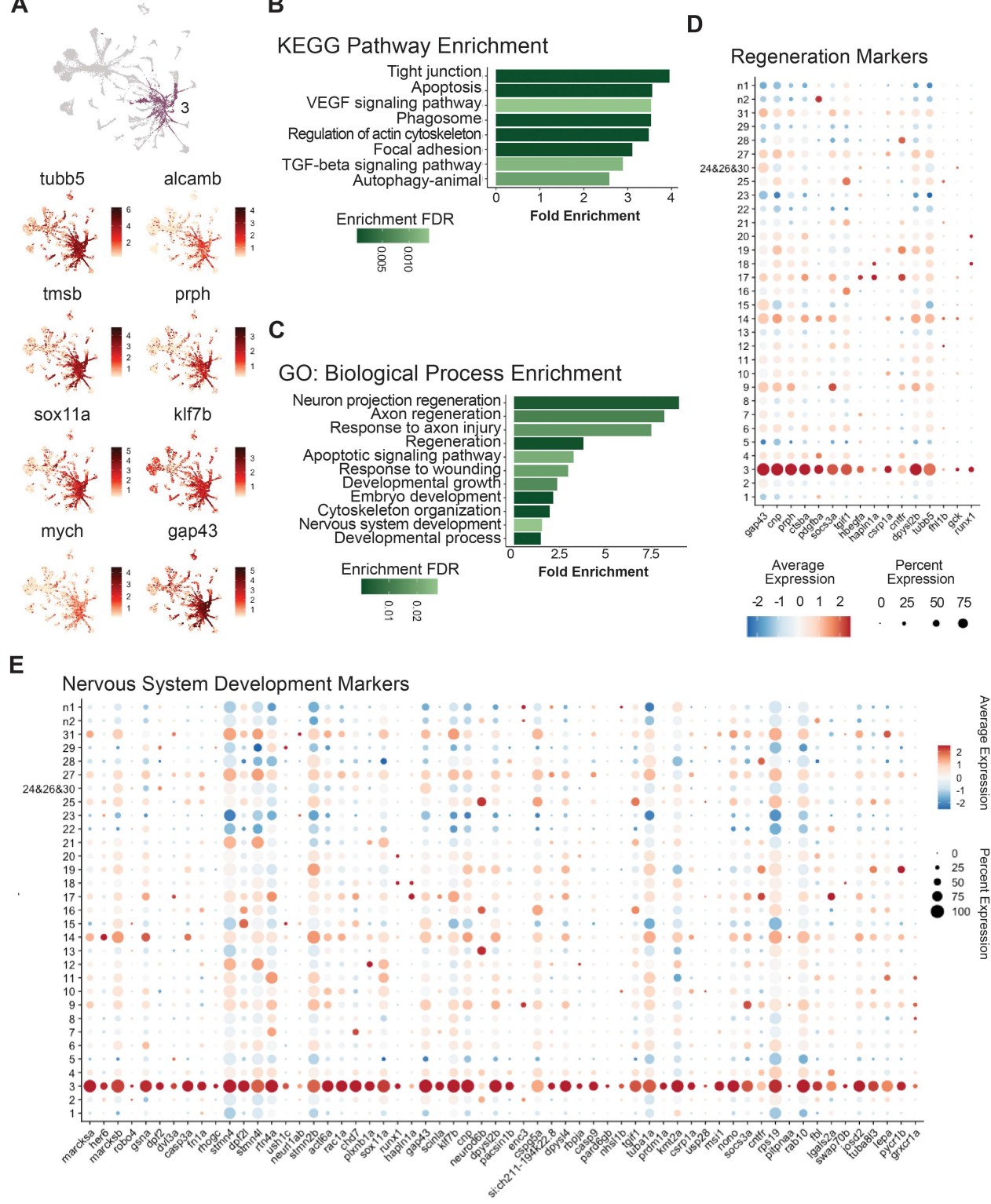

**Fig 4. RGC subtype 3 possesses a progenitor and regenerative identity.** A) Feature plots of differentially expressed genes enriched in the subtype 3 cluster. UMAP with subtype 3 highlighted is included at the top for reference. All feature plots have a min.cutoff of q25. B,C) Curated list of KEGG pathway and GO Biological Process enrichment analysis of subtype 3 retinal ganglion cells. The top differentially expressed genes greater than a log

fold change of 1 and a P value < 0.001 were used for analysis in ShinyGO 0.80 [131]. Bar graphs show fold enrichment of each term and color shade shows enrichment false discovery rate. D,E) Dot plots of all statistically upregulated differentially expressed genes in RGC subtype 3 that fall within the Regeneration Biological Process GO term and the Nervous System Development Biological Process GO term, respectively. Dotplots confirm that the increase in Subtype 3 RGCs found after injury is associated with increased expression of a suite of genes required for nervous system development and regenerative processes.

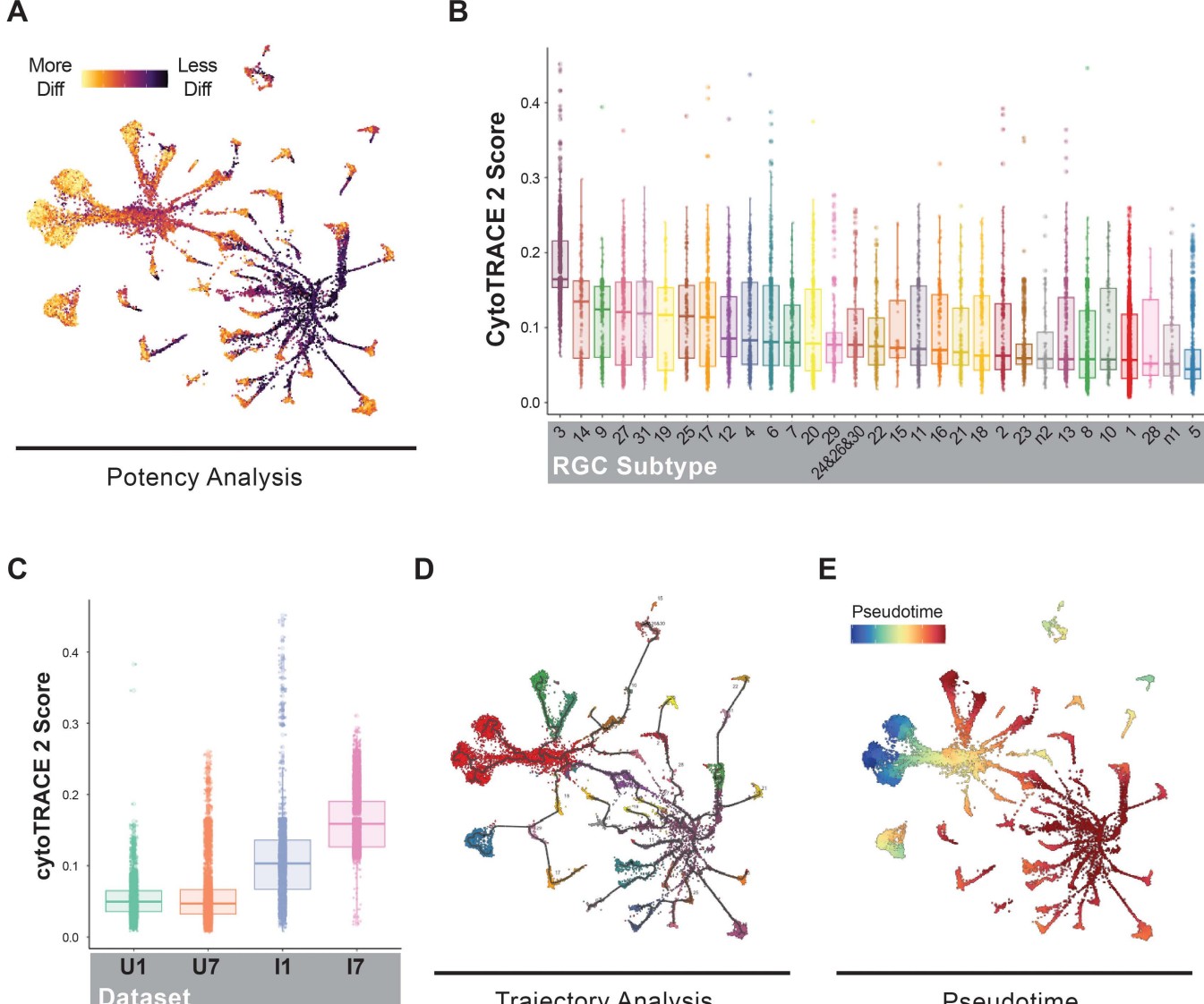

**Fig 5. RGCs dedifferentiate in response to optic nerve injury.** A) UMAP representation of cell potency assessment by CytoTRACE 2 [48]. Darker purple color represents a less differentiated cell state. B) Boxplot representation of CytoTRACE 2 score for each retinal ganglion cell cluster. Higher scores represent a less differentiated state. Cluster 3 shows the highest CytoTRACE 2 score. C) Boxplot representation of CytoTRACE 2 score for each dataset, Uninjured Day 1, Uninjured Day 7, Injured Day 1, Injured Day 7. Both injured datasets show a higher CytoTRACE 2 score suggesting an injury response resulting in a less differentiated state. D) Cellular trajectory graph construction using Monocle3 shows relationships within and between RGC clusters [49]. E) Pseudotime analysis using Monocle3 with cluster 3 set as the root, highlighting potential dedifferentiation trajectories in response to injury.

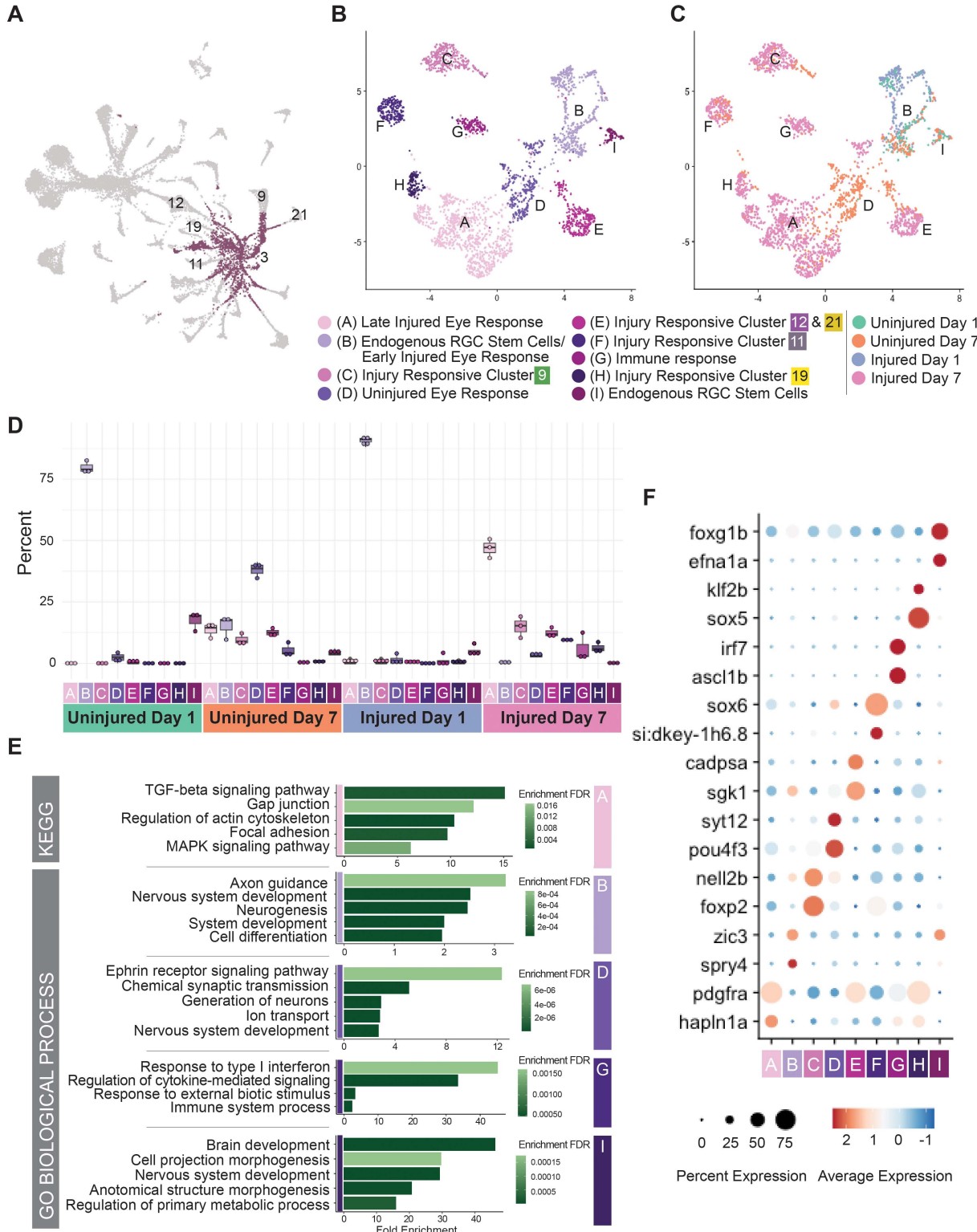

Fig 6. Subtype 3 RGCs are intrinsically heterogeneous and contain immature and injury-responsive subpopulations. A) UMAP representation of all datasets, highlighting RGC subtype 3 in purple. B,C) UMAP reclustering of subtype 3. B) 9 RGC subtype 3 subclusters were identified. C) Dataset identity of subclusters: Uninjured day 1, Uninjured day 7, Injured day 1, and Injured day 7. D) Cluster occupancy relative to dataset highlights

an Uninjured Day 7 response that mirrors an Injured Day 7 response. E) GO Biological Process enrichment analysis using ShinyGO 0.80 of subclusters. Subclusters C, E, F, and H did not show any statistically significant enrichment. F) Dotplot of differentially expressed genes in each subcluster.

expression with other RGC subtype populations (Fig 6F) and are predicted to be close in pseudotime trajectory (Fig 5D and 5E). These subclusters, occupied by Day 7 cells, have been defined by differential gene expression to be subtype 3 RGCs, but they share marker expression with other subtypes, suggesting that these may be subtype cells that are responding to injury. This response could be a transient transcriptional state as cells dedifferentiate into late-responsive subcluster A cells or as they redifferentiate into other RGC subtypes. Alternatively, these cells may never lose their endogenous subtype marker expression in parallel to displaying a distinct transcriptional response to injury. Taken together, these analyses identify heterogeneity within RGC subtype 3 and, surprisingly, that Uninjured Day 7 RGCs possess a remarkably similar expression profile as Injured Day 7 RGCs. This further supports the presence of a systemic response to optic nerve injury, resulting in a similar dedifferentiation process in RGCs of both the injured and uninjured eye.

## There are distinct temporal phases of the RGC injury response

The occupancy of specific subtype 3 subclusters by cells from distinct experimental conditions led us to be interested in understanding bulk transcriptional similarities and differences between each experimental condition. To achieve this, we first combined scRNA-Seq data from all RGC subtypes in each experimental condition to perform pseudobulk analyses and identify differentially expressed genes across conditions (Fig 7 and S4 Table). Pseudobulk analyses use single-cell sequencing data to determine differential expression by aggregating cells into larger, biological sample-level observations. Heatmap and dotplot representations of differentially expressed genes showed that uninjured datasets were more similar to each other than to either the Injured Day 1 or Injured Day 7 datasets (Fig 7A and 7B). In addition, expression profiles of the Injured Day 1 cells were distinct from the Injured Day 7 cells, indicating that there are early and late phases to the response to optic nerve injury. Early response genes included *mych*, *atf3* and *jun*, while late-response genes included *tubb5*, *tmsb* and *alcamb* and *prph* (Fig 7B). GO analysis of Injured Day 1 versus Uninjured Day 1 showed enrichment of wound healing processes at Injured Day 1, as well as the Jak-STAT pathway, which is known to be involved in zebrafish RGC survival (Fig 7C) [14]. By comparison, Injured Day 7 versus Uninjured Day 1 GO analysis showed an enrichment of cytoskeletal changes, potentially in response to axon regrowth, which occurs between 2 and 40 dpi in zebrafish [8,9]. Late response GO enrichment also included cholesterol biosynthesis, endopeptidase activity, DNA methylation, immune responses, autophagy, regulation of the actin cytoskeleton, proteasome activity and metabolic changes. Some of these processes and/or genes associated with them have been implicated in RGC regeneration and resiliency responses [50–59], while others are relatively unstudied in neuroprotection and regeneration. Looking specifically at genes within the immune, autophagy and regulation of the actin cytoskeleton pathways revealed differential gene expression, with temporal specificity, during the injury response (Fig 7E–7G). For example, the immune response included an early phase where genes like *stat1a* and *stat3* were enriched, and a late phase that included genes like *socs3a*, *socs3b* and *ifi45* (Fig 7E). Some immune genes also were expressed early and sustained through later time points and these included *jak1*, *irf2* and *irf9*. Similarly dynamic expression was detected for autophagy related genes (Fig 7F) and actin cytoskeleton regulator genes (Fig 7G). We also generated bulk RNA-seq data to support our pseudobulk analysis and confirmed the time-dependent response to injury (Fig 7H). Bulk analyses included RGC samples collected at 5dpi and their transcriptome was similar to 7dpi samples. This suggested that the transition from early phase to late phase responses occurs before 5dpi (Fig 7H).

## The uninjured eye response to injury is isolated to specific transcriptional targets

Our analysis of RGC subtype 3 heterogeneity revealed subpopulations of subtype 3 RGCs that were occupied specifically by Injured Day 7 cells and Uninjured Day 7 cells suggesting that there may be a shared systemic response to optic

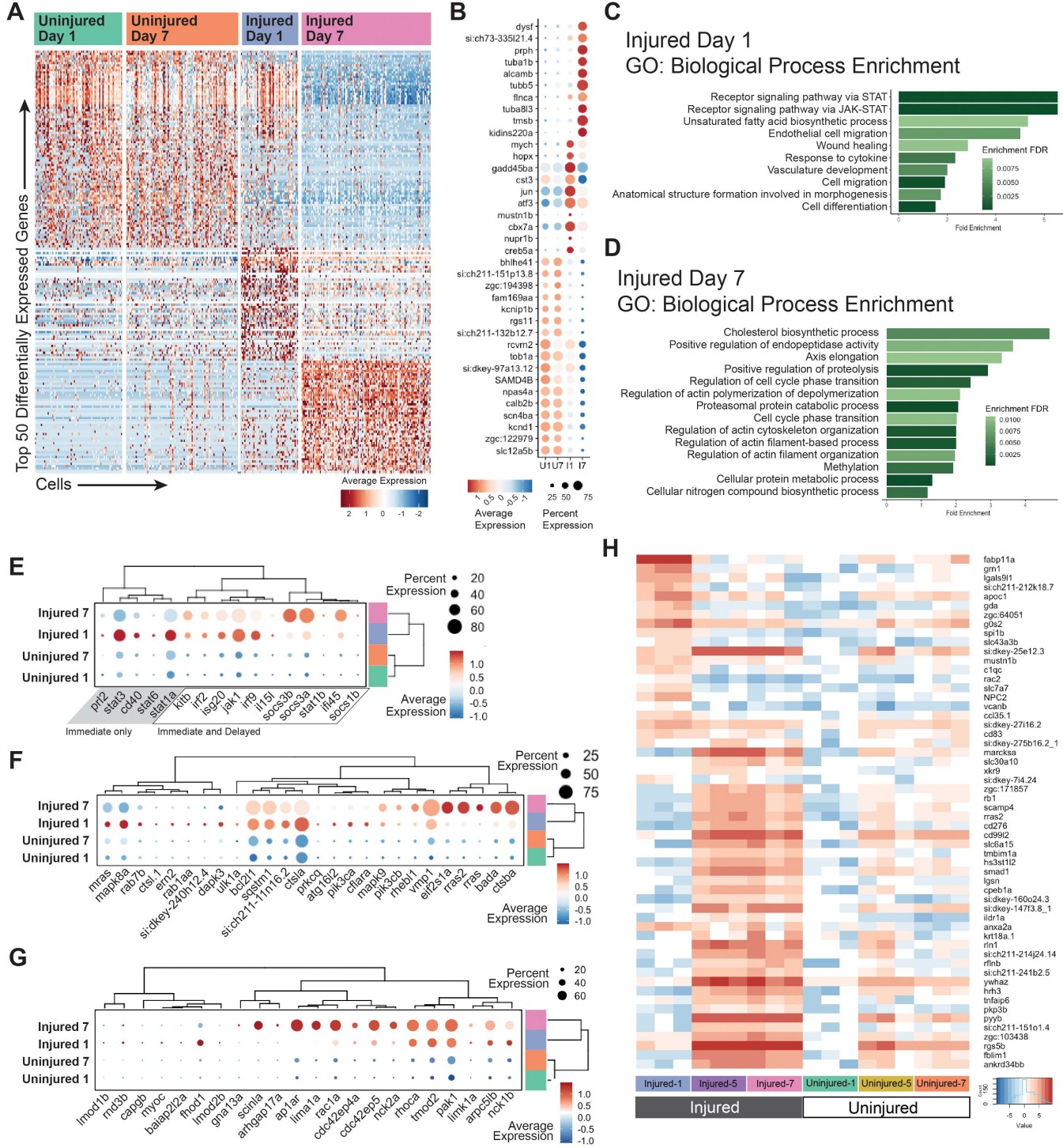

**Fig 7. There are distinct temporal phases of the RGC injury response.** A) Heatmap of the top 50 differentially expressed genes in pseudobulk analysis of scRNA-Seq data from all experimental conditions, including Uninjured Day 1, Uninjured Day 7, Injured Day 1 and Injured Day 7. B) Dotplot of the top 10 differentially expressed in pseudobulk analysis of experimental conditions. C,D) Curated list of unique terms from GO Biological Process enrichment analysis of both Injured Day 1 and Injured Day 7. Early injury response highlights immune response and wound healing. Later injury response highlights axon regrowth processes. E) Dotplot of immune response genes shows differences in early and late immune-related gene expression F) Dotplot of autophagy response genes G) Dotplot of genes associated with the regulation of actin cytoskeleton. H) Heatmap a bulk RNA-seq timecourse experiment. Samples were collected in triplicate of Injured Day 1, Injured Day 5 and Injured Day 7, and the corresponding Uninjured eye. The heatmap is composed of the top differentially expressed genes in the injured datasets.

nerve transection displayed in the contralateral Uninjured Day 7 eye. To better understand this uninjured-eye response and determine whether it is similar to that in the Injured eye, we performed GO and KEGG analyses of Injured Day 1, Injured Day 7 or Uninjured Day 7 versus Uninjured Day 1 pseudobulk datasets, focusing on shared enrichment between the datasets (Fig 8A and 8B). Uninjured Day 7 shared the enriched terms 'developmental processes', 'immune response', 'phagocytosis' and 'response to wounding', among others (Fig 8A and 8B). We next identified the top differentially expressed genes between Uninjured Day 1 and Uninjured Day 7 RGCs and showed that none of these genes are specifically enriched in the uninjured eye; rather, their expression mirrors that in the injured eye. (Fig 8C). Feature plots show that these genes are upregulated throughout the Injured Day 7 cells, no matter to what RGC subtype they are assigned (Figs 8D and S8). Expression was variably similar in Injured Day 1 RGCs, suggesting that this is a temporally-dependent response. We detected a similar upregulation of these genes in Uninjured Day 7 RGCs, although less robust and centered in the RGC subtype 3 population. Taken together, these data show that RGCs in the contralateral eye respond to optic nerve injury by mounting a mirrored response that is molecularly similar to that detected in injured RGCs, despite not being directly injured themselves.

### RNA-FISH confirms an increase in subtype 3 marker gene expression *in vivo* in RGCs of both the injured and uninjured contralateral eye after injury

Single cell profiling of RGCs after optic nerve transection suggested an increase in RGC subtype 3 gene expression in both the injured and uninjured eye and in a time dependent manner. To validate these data *in vivo*, we quantified the expression of the subtype 3 RGC marker genes *tubb5* and *alcamb* [25] using hybridization chain reaction (HCR) based RNA fluorescence *in situ* hybridization (RNA-FISH). Imaging of flat-mounted whole retinae demonstrated that while *tubb5* and *alcamb* were not detected in the naive retina (Fig 9C), both genes were robustly expressed in RGCs across the Injured Day 7 retina (Fig 9A). Moreover, RNA-FISH confirmed expression in the Uninjured Day 7 retina (Fig 9B). Interestingly, expression in the Uninjured Day 7 retina was localized to concentric circles of RGCs emanating from the optic disc. There was variability in the number of expression zones detected (i.e., the number of circles) and the continuity of these circles of expression between retinae (e.g., Fig 9B'); however, all Uninjured Day 7 retinae examined showed this pattern.

We next quantified*tubb5* and *alcamb* expression throughout the injury response by assessing expression at days 1, 3, 5 and 7 post-injury, utilizing an unbiased sampling of four peripheral and four central regions per retina for quantification. Expression was significantly elevated in Injured Day 1 (*tubb5* only) and in Injured Day 3, 5 and 7 for both markers when compared to all other experimental conditions as well as to naive controls (Figs 10 and S9). Unbiased high magnification sampling detected some retinal regions of clustered RGC expression while RGCs in other retinal regions were devoid of expression, consistent with the patterned distribution of expression detected in the whole retina, especially in the Uninjured datasets (Fig 10).

We were also curious if the dedifferentiation response and shift towards RGC subtype 3 gene expression was transient post-injury or whether it was sustained at 25 or 50 dpi, after which axon regeneration is thought to be complete [8,9]. RNA-FISH of injured retinae at 25 and 50 dpi showed no expression of *tubb5* and *alcamb* indicating that upregulation of RGC subtype 3 genes in injured RGCs is in fact transient and returns to baseline levels, coincident with RGCs recovering their *isl2b*:GFP signal (S10 Fig). Taken together, these data confirm that RGCs adopt a subtype 3 identity post-optic nerve transection but that they do so transiently. Moreover, these data also confirm that the injury response manifests in the uninjured eye at 7dpi, further demonstrating the systemic response to optic nerve injury.

## Discussion

Unlike in mammals, RGCs in the zebrafish retina survive severe optic nerve injury. The mechanisms underlying this intrinsic neuroprotective ability remain unknown, but could be transformative in clinical treatment of glaucoma, optic neuropathy and ocular trauma patients if they could be leveraged in humans to preserve RGCs after injury or in the diseased eye.

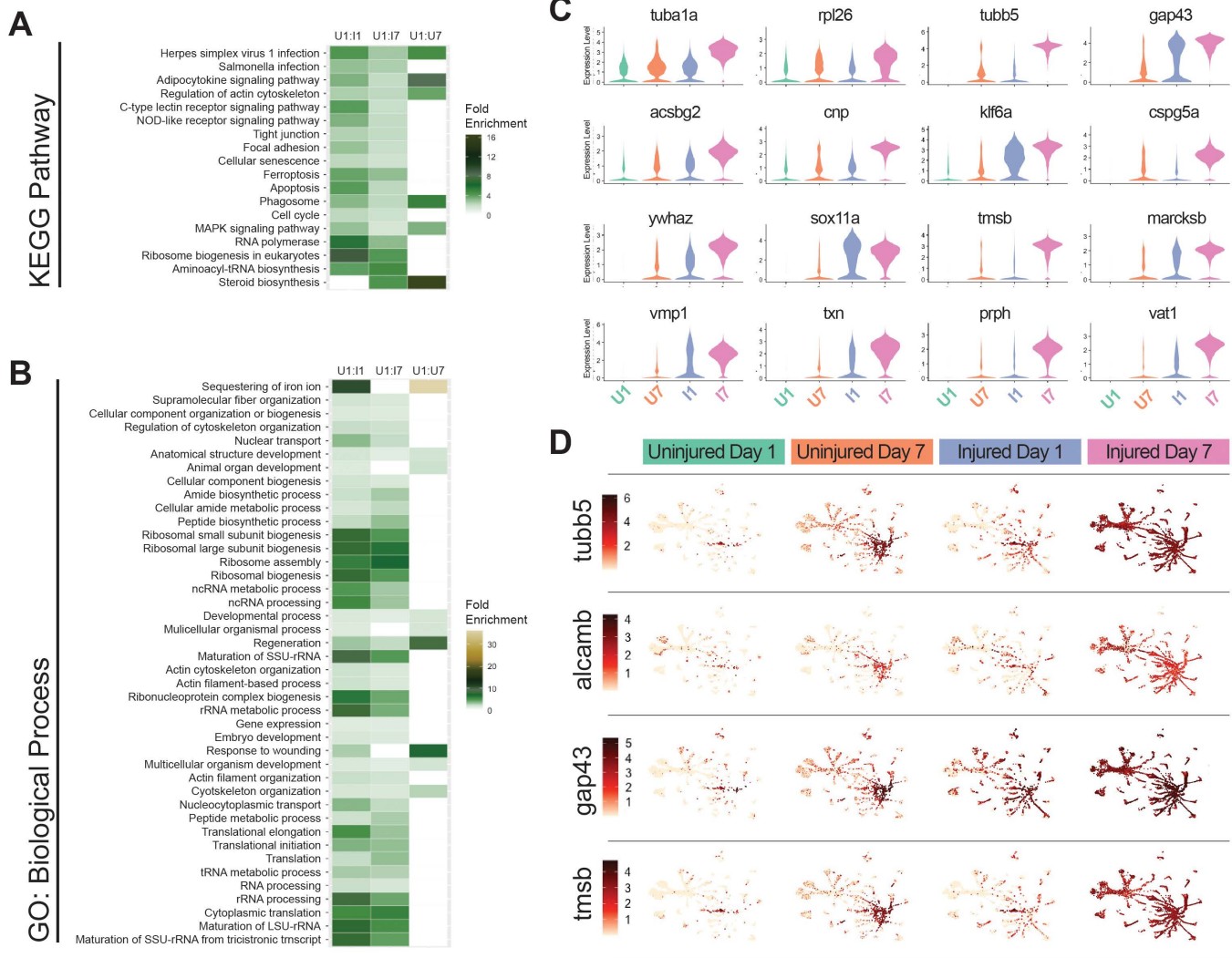

**Fig 8. The uninjured eye response to injury is isolated to specific transcriptional targets.** A,B) Heatmap of KEGG & GO Biological Process enrichment analysis of shared terms across experimental conditions relative to Uninjured Day 1. All fold enrichment included has a false discovery rate < 0.01. This analysis shows a clear uninjured eye response at Day 7 that reflects an injured eye response. C) Violin plots of genes that show statistically significant differential expression in the Uninjured Day 7 eye relative to Uninjured Day 1 eye. This suggests that some genes are primed for transcriptional activation in response to immune response. D) Feature plots of genes *tubb5, alcamb, gap43* and *tmsb* split by their experimental condition, Uninjured Day 1, Uninjured Day 7, Injured Day 1, and Injured Day 7.

Here, we utilized scRNA-Seq to identify intrinsic neuroprotective mechanisms in zebrafish and characterize the molecular underpinnings of RGC resilience to injury. Our results revealed that: 1) all RGC subtypes are resilient to injury in zebrafish; 2) zebrafish RGCs dedifferentiate in response to injury and adopt a molecular phenotype resembling RGC subtype 3, an endogenous immature RGC subtype found in the uninjured, wildtype adult eye; 3) there are distinct early and late phases to the RGC injury response; and 4) specific sets of genes in the contralateral uninjured eye show similar transcriptional responses to that in the injured eye. These results are exciting as they provide a resource for further understanding the cell-specific regulation of intrinsic neuroprotection and identify numerous putative neuroprotective genes and pathways in RGCs that can be further interrogated. The concept that RGCs change their cell state in response to injury is exciting

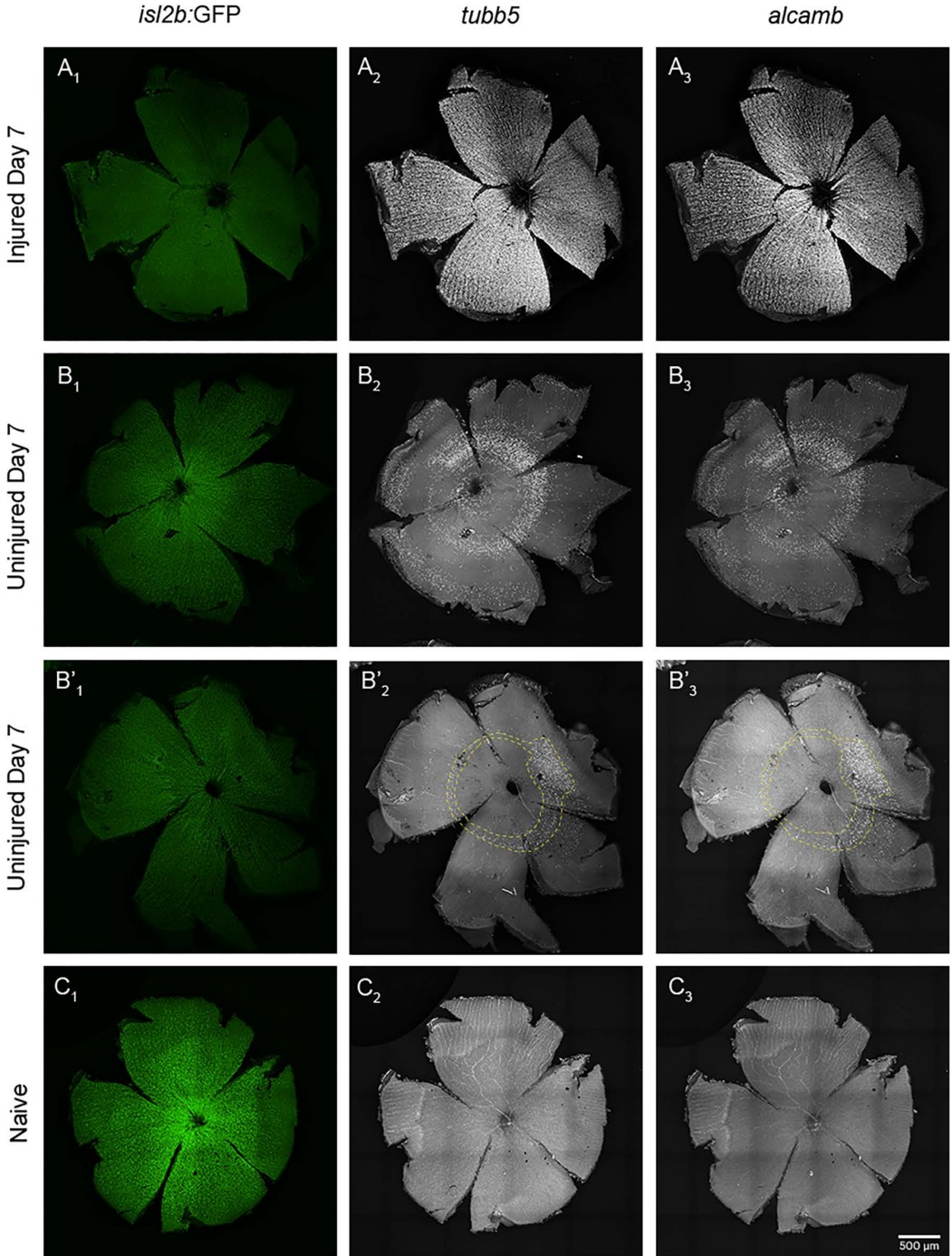

*isl2b*:GFP     *tubb5*     *alcamb*

**Fig 9. RNA-FISH confirms an increase in subtype 3 marker gene expression *in vivo* in RGCs of both the injured and uninjured contralateral eye after injury.** Representative images of the whole retina at 7 days post ONT or naive (4X objective and stitched together, n=6 per condition) show that A) RGCs at Injured Day 7 ($A_1$) robustly express *tubb5* ($A_2$) and *alcamb* ($A_3$). B - B') RGCs at Uninjured Day 7 ($B_1$ and $B'_1$) express *tubb5* ($B_2$ and $B'_2$) and *alcamb* ($B_3$ and $B'_3$) which are localized to concentric circles of RGCs emanating from the optic disc, albeit with some variability in the magnitude of these concentric circles of expression. C) RGCs of naive retina ($C_1$) do not express *tubb5* ($C_2$) and *alcamb* ($C_3$).

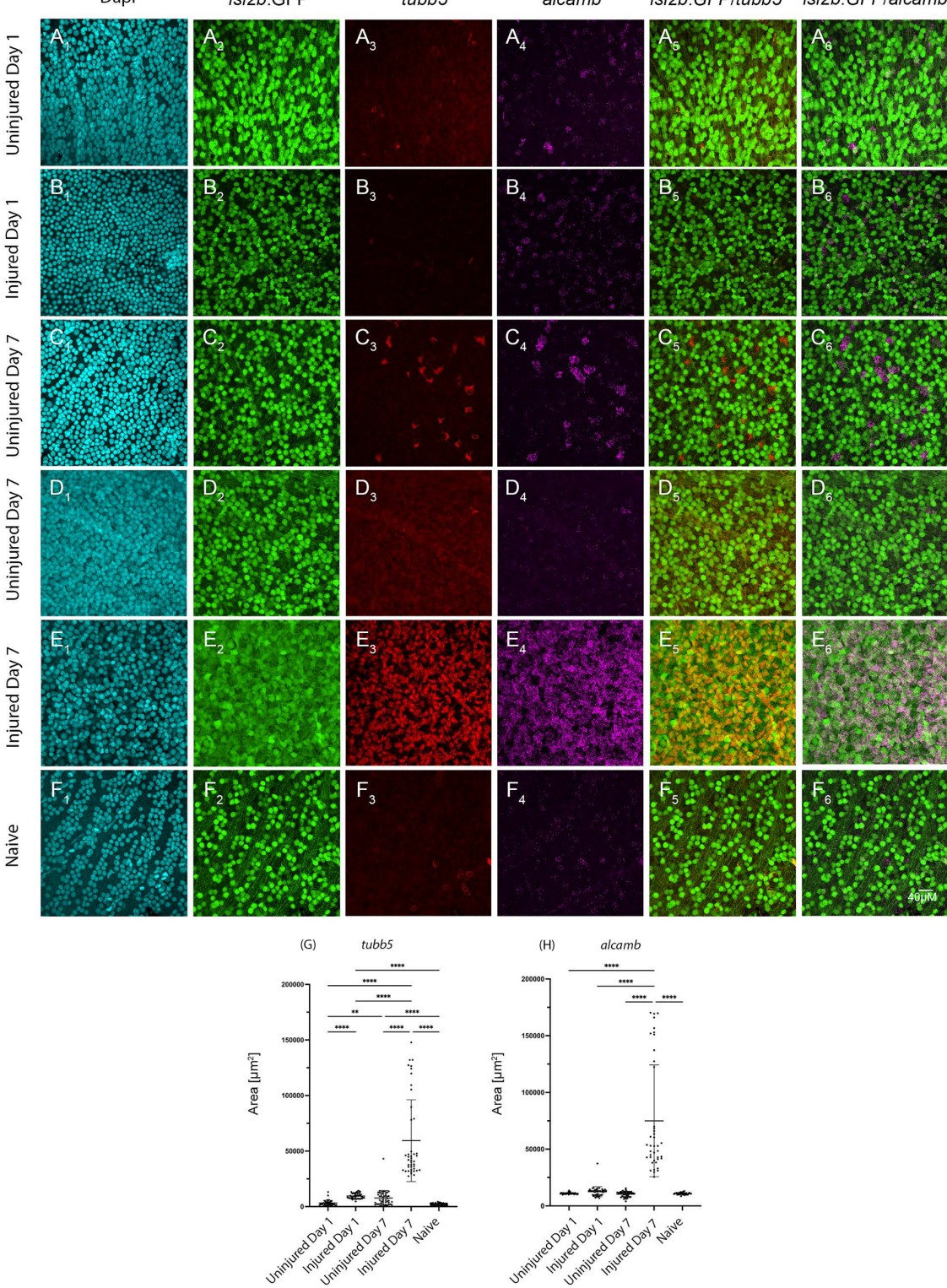

**Fig 10. Quantification of subtype 3 marker gene expression reveals significant differences in response to ONT.** Representative images of high resolution fields of view at 1 and 7 days post ONT or naive retina (60X objective, n = 24 fields of view per condition) show that A) Uninjured RGCs at 1 dpi show minimal/baseline levels of immature gene expression ($A_3$-$A_4$), similar to *tubb5* and *alcamb* expression in naive retina.B) Injured retina at 1 dpi shows some expression of immature genes ($B_3$-$B_4$). However, C) uninjured RGCs at 7 dpi show robust expression of immature genes in clusters of

RGCs in some regions ($C_3$- $C_4$) and not others ($D_3$-$D_4$). High resolution imaging confirms robust and uniform expression of immature genes *tubb5* and *alcamb* ($E_3$ - $E_4$) in injured retina at 7dpi. F) Naive retina showing baseline levels of immature gene expression ($F_3$-$F_4$). G) *tubb5* expression increases significantly after ONT at both 1 and 7 dpi compared to respective uninjured and naive retina. *tubb5* expression in uninjured retina at 1 dpi is not significantly different from naive retina, however, it is highest at 7 dpi. H) *alcamb* expression is significant only at 7dpi compared to injured eyes at 1 dpi, uninjured eyes, and naive controls.

and warrants further study as a potential approach in mammalian systems to preserve damaged RGCs and/or stimulate regenerative responses when RGC axons have been damaged. Finally, our results also indicate that caution should be taken in experimental optic nerve injury studies where the contralateral eye is used as a control given that there is a robust injury response elicited by contralateral uninjured RGCs in response to a unilateral optic nerve injury.

## All RGC subtypes are resilient to injury in zebrafish

RGCs are heterogeneous in all animals examined, including humans [24,60]. RGC subtypes in animal models have been classified by morphology (e.g., [61]), molecular features (e.g., [62,63]) or activity (e.g., [64–66]), while more recently, scRNA-Seq data have further refined RGC subtype predictions (e.g., [21–23,25]). Utilizing a variety of models and injury paradigms, it has been shown that there is differential resilience and susceptibility of RGC subtypes to injury in mammals (reviewed in [67–69]). Our data show that all subtypes of RGCs in zebrafish survive after optic nerve transection, and thus, that there is no differential resilience and susceptibility of RGC subtypes in zebrafish. Moreover, we also determined that mostRGCs survive optic nerve injury in zebrafish. This observation is in contrast to previous work by us [14] and others [10] indicating a small, ~20% die off of RGCs post injury. We did detect a decrease in RGC density within the ganglion cell layer, indicating that some RGCs are lost post-injury, but likely not at the level we previously thought based on *isl2b*:GFP expression. Indeed, proliferative Muller glia, the source of regenerated neurons in zebrafish (reviewed in [70]), are not detected in the optic nerve transected retina as would have been expected if large numbers of RGCs were dying and needed to be replaced [10,14]. The lack of substantial RGC death is also consistent with results in goldfish after optic nerve crush, where almost no death of RGCs has been detected [71]. Axonal regeneration and visual function recovery after optic nerve injury are rapid in zebrafish, suggesting a model in which the preservation of existing RGCs is critical for the rapidity of regeneration and recovery [8,9,47].

## RGCs alter their cell state and dedifferentiate in response to injury

While few TUNEL+ RGCs were detected in the injured retina, differential abundance quantification revealed an increase in subtype 3 RGCs and concomitant decreases in several other RGC subtypes after injury, with pronounced subtype 3 increases at 7dpi. RGC subtype 3 was previously identified by scRNA-Seq as a population of immature, progenitor-like RGCs that remain in the adult retina [25]. CytoTRACE2 analyses demonstrated that subtype 3 RGCs, as well as those RGC subtypes with close predicted pseudotime relationships to subtype 3 after injury, are the least differentiated RGCs in the injured eye. Gene expression changes associated with subtype 3 and the dedifferentiation response included upregulation of *tubb5*, *alcamb*, and *tmsb*, as well as numerous genes associated with regeneration and reprogramming in other systems. For two of these genes, *tubb5* and *alcamb*, we verified upregulation *in vivo* after injury in RGCs across the retina. By 25dpi, expression of these immature markers is no longer detectable, indicating that the dedifferentiated phenotype is not sustained. Interestingly, this timing correlates well with the period during which axon regeneration is occurring in zebrafish, which is thought to be complete by 40dpi [8,9]. These data strongly support a model in which RGCs dedifferentiate in response to injury, likely as a means to ensure survival and enable axon regeneration and then return to a wildtype transcriptional profile after optic nerve regeneration is complete.

Dedifferentiation as an injury response to promote tissue regeneration is well documented in various somatic organs and tissues, such as the retina [72–74], limb [75,76], heart [76–78] and intestine [79]. In these paradigms, however,

dedifferentiation involves cellular reprogramming to generate progenitor cells that proliferate and regenerate the lost tissue. This is not what we're observing in zebrafish RGCs after optic nerve transection, as there is no need to repopulate the RGC pool; instead, there is a need to keep the RGCs alive and plastic, such that they can regenerate their axons and restore visual function. The molecular mechanisms underlying dedifferentiation of zebrafish RGCs for neuroprotection could be mediated by conventional and/or novel combinations of upregulated reprogramming-related transcription factors such as sox, klf and others. Indeed, adult mammalian RGCs can be reprogrammed into a younger state by overexpression of Oct4, Sox2, and Klf4, leading to protection of RGCs after optic nerve injury [89] or in an inflammation-mediated ocular degeneration model [80]. *oct4*, *sox2* and *klf4* are upregulated in zebrafish RGCs shortly after injury [28], suggesting that these could play a role in the dedifferentiation response.

The ability of neurons to dedifferentiate and/or adopt immature phenotypes after injury to facilitate survival and/or regeneration responses has recently been reported in several systems (reviewed in [81]). For example, in the mouse retina, adult RGC subtypes have been identified that retain signatures of embryonic gene expression and while these RGC subtypes do not regenerate intrinsically after injury, they are more regenerative than their more differentiated counterparts in a *Pten* knock-down model [27]. Corticospinal motor neurons in mice respond to injury by reverting to an embryonic gene expression program which is necessary for them to regenerate [82]. Likewise, dorsal root ganglion neurons reprogram after sciatic nerve crush and lose subtype specific identities during the injury and axon regeneration responses [83]. In neurons, and particularly RGCs, we speculate that this dedifferentiation response enables neurons to change their cell state, thereby activating or maintaining pro-survival pathways, which provide sufficient time for axonal regeneration to occur and neuronal circuitry to be reestablished.

A major question remains as to the function(s) of the small population of subtype 3 RGCs in the uninjured adult eye. Our in-depth profiling of gene expression within subtype 3 RGCs revealed subpopulations of endogenous, progenitor-like cells as well as injury-responsive subpopulations. Given that RGCs are post-mitotic and do not proliferate *in vivo*, the role of these progenitor-like subtype 3 RGCs in the normal/uninjured retina remains unclear. It will be of interest to generate transgenic lines to label these cells in both the uninjured and injured retina, to enable tracking of their axonal projections as well functional manipulations, to begin to address this.

### There are distinct early and late phases to the injury response

Molecularly, the RGC injury response in zebrafish can be divided into distinct early and late phases. Early response genes included *mych*, *atf3* and *jun*. GO analyses also showed enrichment of the Jak-STAT pathway, wound healing, and several other processes during the early injury response phase. Jak/STAT is known to contribute to RGC survival and regenerative responses (e.g., [14,16,84–88]). Likewise, several of the other early response genes identified also possess known functions related to neuronal survival and regeneration. For example, c-myc is downregulated in mouse RGCs after optic nerve injury, but ectopic expression promotes survival and axon regeneration post-injury [89]. Myc paralogues are known to regulate Muller glia reprogramming in response to retinal injury in zebrafish [45,90], but their role in RGC survival and axon regeneration after optic nerve injury in fish is unknown. In mice, ATF3 is upregulated early in the response to optic nerve crush and is thought to regulate the expression of immune and inflammatory signaling cascades, possibly contributing to the death of RGCs [91]. There are also reports, however, that ATF3 overexpression can preserve RGCs in mouse after damage [92], while in zebrafish spinal cord regeneration, atf3 is required for axon regeneration and may function to limit pro-inflammatory signaling [93]. These latter, pro-survival and pro-regenerative roles are consistent with observations in other neural contexts, where ATF3 is part of a core set of pro-survival and regenerative transcriptional regulators (e.g., [83,94,95]; reviewed in [96]).

Examination of immune system-related gene expression also revealed a biphasic immune response after optic nerve injury, with early and late components. The immune system is a known regulator of cell survival and cell death in a variety of neural cell types and tissues, including RGCs. It is thought that modulation of immune responses is required for

successful cell survival and regeneration within these cells and tissues (reviewed in [50]). Early injury phase immune response genes included *stat1a*, *stat3* and *cd40*. Stat1 and Stat3 play roles in facilitating RGC survival after injury and during axon regeneration in mice [88,97,98], while stat1a and CD40 both play roles in Muller glia after retinal injury or during the progression of diabetic retinopathy, respectively, where they are thought modulate inflammatory responses of microglia and macrophages in response to injury [99,100]. We hypothesize that injured RGCs likewise upregulate *stat1a* and *cd40* to modulate immune cell activity and promote RGC survival and ultimately, axon regeneration. Late injury phase immune response genes included immune-related genes such as *socs3a* and *socs3b*, well known modulators of RGC axon regeneration [16,101,102] and *ifi45*, an interferon-induced gene in zebrafish [103].

Finally, we identified late phase injury response genes and pathways and again, several stood out with known roles in RGC survival and optic nerve regeneration. For example, we identified genes involved in endopeptidase activity, which is known to contribute to RGC axon regeneration in mice [57], possibly downstream of ATF3 activity [104]. *anxa2a*, a zebrafish Anxa2 paralog, was upregulated post-injury and is known to protect RGCs and enhance RGC axon regeneration in a mouse optic nerve crush model [105]. Likewise, genes involved in DNA methylation were also upregulated as part of the late phase injury response, and changes in methylation are thought to contribute to RGC reprogramming and axon regeneration (e.g., [58,59,106,107]). For all of the early and late phase genes identified, it will be of interest to characterize their intrinsic neuroprotective roles as they may represent viable targets or pathways to pursue in mammals and, ultimately, as targets for development as therapeutics for use in humans to protect RGCs after injury or in the diseased eye.

### The contralateral eye response

Our scRNA-Seq data revealed a robust transcriptional response in RGCs of the contralateral uninjured eye at 7dpi. Gene expression changes in the uninjured RGCs mirrored those in injured RGCs, but with a variably lower magnitude of response. Indeed, this uninjured eye response included RGC markers for dedifferentiation and a shift towards subtype 3 identity. We validated gene expression changes *in vivo* and showed that an injury-like molecular response occurred in the uninjured eye. These data demonstrate that RGCs of the contralateral eye respond to injury similarly to those in the injured eye, despite not being directly injured themselves. Interestingly, changes in gene expression in the uninjured eye presented spatially as concentric circles, emanating from the optic disc. We do not know the significance of this pattern but hypothesize that it may have some relationship to the topography of RGC axon bundling in the optic nerve and possible modes through which the injury signal is conveyed between the injured and uninjured eyes [108–112].

A contralateral eye response to unilateral optic nerve injury has been reported in animal models as well as human patients (reviewed in [113]). However, the impact of the contralateral response in modulating cell survival has been debated, with some studies indicating that RGCs of the contralateral mouse retina die after unilateral optic nerve injury (e.g., [114]) and others indicating that contralateral eye RGC death does not occur (e.g., [115]). Regardless of the impact on RGC survival, our results highlight robust molecular responses of RGCs in the contralateral eye of zebrafish after unilateral optic nerve transection, indicating that the uninjured RGCs do respond to injury. In the zebrafish, where most RGCs survive after optic nerve injury, there is no resultant RGC death in the contralateral eye, but gene expression changes in affected RGCs are pronounced. It is almost certain that some aspects of the contralateral eye response in zebrafish are mediated by activation of the immune system, as has been observed in other systems. For example, in rat models of unilateral ocular hypertensive glaucoma [116], optic nerve crush [117] or optic nerve transection [118], significant microglia activation is detected in the contralateral eye after injury. Broadly inhibiting microglia activation or inflammation limits contralateral RGC responses [119]. Astrocytes may also be involved in this systemic response to unilateral injury. Astrocytes of the contralateral eye become reactive after unilateral optic nerve crush in mice [120]. Moreover, a recent study showed that astrocytes from a healthy eye can transfer metabolites to the injured eye when it is damaged, but in doing so become more vulnerable to stress and ultimately degenerate [121]. It will be of interest to characterize these responses in zebrafish and leverage the system to better understand the physiological and molecular components of the contralateral eye response.

## Materials and methods

### Ethics statement

All animals were treated in accordance with provisions established by the University of Pittsburgh School of Medicine or University of Texas at Austin Institutional Animal Care and Use Committees (IACUC) and all protocols were approved by these entities.

### Zebrafish husbandry

*isl2b*:GFP [122] transgenic zebrafish (*Danio rerio*) used in this study were 3–6 months old adults, maintained on a 14:10 hour light: dark cycle at $28^0$C. Previous work from our lab has shown that there was no difference in RGC survival based on sex of the fish [14]; therefore, male and female were utilized for experiments in this study.

### Optic nerve transection (ONT)

ONT was performed as previously described [10,14,16] under a dissecting scope (Leica E65S). The optic nerve in the left eye was injured while the right (contralateral) eye was the uninjured sham experiment. Tissue from around the eye was dissected away, the orbit rotated, and the optic nerve transected using forceps. Sham experiments followed the same procedure, without optic nerve transection. Care was taken not to damage the ophthalmic artery running alongside the nerve. On the rare occasion that the artery was cut, the animal was euthanized.

### NMDA induced lesions

1uL of 100 mM NMDA or PBS (control) was injected into the left eye, as described previously [123], to be used as a positive control for RGC apoptosis [35]. Briefly, fish were anesthetized in 0.02% tricaine methanesulfonate and under microscopic visualization, a small incision was made in the cornea with a 30G needle. A Hamilton syringe with a blunt 30G needle was inserted through this incision to deliver the NMDA solution or PBS behind the lens.

### RGC purification and droplet based single cell RNA sequencing

RGCs in transgenic Tg(*isl2b*:GFP) zebrafish express GFP [122] and were used to isolate RGCs during fluorescence activated cell sorting (FACS). To do this, retinas from adult fish were dissected in oxygenated (ox) Ames and transferred into ox Ames on ice until tissue collection was completed. 20 retinas per sample (injured and uninjured at 1 and 7 dpi) were dissected and dissociated per experiment. Retinas were digested in a solution containing papain (20U/ml), DNase I (80U/ml), and L-cysteine (1.5mM) in ox Ames at $28^0$C for 45 minutes. To stop the digestion, the papain solution was replaced by papain inhibitor solution containing ovomucoid (15mg/ml), BSA (15mg/ml) and DNase I (80U/ml). Tissue was gently dissociated in the inhibitor solution by triturating with a flamed glass pipette. The cells were washed by pelleting at 250 g for 8 minutes and resuspending in ox Ames containing 0.4% BSA. The cell suspension was filtered through a 30μm strainer prior to FACS purification. Non-transgenic wild-type retinas were used to determine background fluorescence levels and adjust sorting gates. Propidium iodide was added to distinguish live GFP$^+$ RGCs. Cells were washed and resuspended in PBS 0.04% BSA and loaded onto the 10X Genomics microfluidic device within ~45 minutes after FACS enrichment. Each sample was isolated and collected in triplicate, with the exception of 1dpi, which was collected in quadruplicate, for biological replicates.

### 3' droplet based single cell sequencing

Single cell libraries were prepared using the single-cell gene expression 3' v2 kit on the Chromium platform (10X Genomics, Pleasanton, CA) following the manufacturer's protocol. Briefly, single cells were partitioned into Gel beads in EMulsion (GEMs) in a 10X Genomics Chromium controller followed by cell lysis and barcoded reverse transcription of RNA,

amplification, enzymatic fragmentation, 50 adaptor attachment and sample indexing. Libraries were sequenced on a single NovaSeq S2-100 flowcell (Paired end reads: Read 1, 26 bases, Read 2, 98 bases).

## Single-cell sequencing data processing

The cellranger count pipeline (version 6.1.2) was used for alignment, producing a filtered matrix containing UMI counts [124]. We used the zebrafish reference genome (Danio rerio.GRCz11.106) and default arguments. Next we used the Seurat package to preprocess and integrate the samples, following the preprocessing workflow outlined by Kölsch et al. (2021) [25] and starting with a total of 104,003 cells [25,125]. Briefly, for each sample, filtered matrices were log-normalized using the NormalizeData function with default arguments. Next all genes were scaled and centered using the ScaleData function and highly variable genes were identified using FindVariableFeatures with nfeatures = 2000. Samples were then integrated using the FindIntegrationAnchors and IntegrateData functions with dims = 1:40. Next, integrated data were re-scaled and PCA, TSNE, and UMAP embeddings were computed using Seurat's RunPCA, RunUMAP, and RunTSNE functions. Lastly, an initial clustering of the integrated data was computed using FindNeighbors (dims = 1:40) and FindClusters.

To focus on RGC populations, clusters that expressed marker genes for contaminant cell types and lacked expression of RGC markers were filtered and removed. Specifically, we removed clusters that had high expression of at least two of the following non-RGC genes: *opn1lw2, opn1mw2, opn1mw1, opn1sw2, opn1lw1, gngt2b, gngt2a, rho, crx, pde6c, pde6ga, opn6a, vsx1, glula, cabp5a, cabp2a, gng13b, gad1b, cd82a, gad2, pax6b, pax6a, cd74a, fcer1gl, and apoeb*. We also annotated and removed doublets from each sample using scDblFinder [126] with default parameters. Our final dataset consisted of 17,769 cells, with many removed *isl2b*:GFP⁺ cells likely being photoreceptors [122]. As part of our quality control assessments, we quantified the percent mitochondrial gene expression and assessed batch effects versus biological variation in our analyses (S3A–S3E Fig).

## Plots

Dimensional reductions, dotplots, violin plots, feature plots, tile plots, box plots, heatmaps, and bar graphs were generated using the Seurat package, ggplot and scCustomize [125,127].

## Label transfer and data integration

32 distinct subtypes of RGC have been previously identified in zebrafish [25]. Transfer of the subtype labels to our data using these published data as a reference was performed using Seurat's label transfer workflow. Briefly, transfer anchors were identified using FindTransferAnchors and then labels transferred using the TransferData function, both with dims = 1:30. Due to technical and biological differences between Injured and Uninjured 7dpi samples and the reference data, we first transferred subtype labels from the reference data to Uninjured 1dpi data and subsequently transferred those subtype labels from Uninjured 1dpi to the remaining samples. We did not identify subtype 32 (the rarest subtype in the reference data) in our samples. Additionally, cells annotated as subtype 24, 26, and 30 clustered together in our data, so we combined those subtypes into one subtype named 24&26&30. We also identified two novel subtypes, which we annotated as n1 and n2. Our data were also integrated with previously published wildtype RGC adult single cell dataset for analysis [25]. Integration was performed by using the Seurat SelectIntegrationFeatures function, the FindIntegrationAnchors function and the IntegrateData function. UMAPs of subclustered subtype 3 cells and Injured and Uninjured datasets were performed by merging the data.

## Differential gene expression analysis

To determine differentially expressed genes (DEGs), we first identified the top 5,000 highly variable genes using scran's modelGeneCV2 followed by getTopHVGs with var.field = "ratio" [1]. We then used findMarkers function to identify the DEGs with direction = 'up', pval.type = 'all', and test.type = "wilcox" [128].

## Differential abundance analysis

Differential Abundance (DA) Analysis was performed two ways, using the edgeR package as described in Orchestrating Single-Cell Analysis and the MiloR method for differential abundance analysis on KNN graph in the Bioconductor vignettes(http://bioconductor.org/books/3.16/OSCA.multisample/differential-abundance.html, https://bioconductor.org/packages/release/bioc/vignettes/miloR/inst/doc/milo_demo.html) [129,130]. We performed DA of subtypes between each condition (Uninjured 1dpi, Uninjured 7dpi, Injured 1dpi, Injured 7dpi) to quantify differences in subtype abundances in relation to injury. First, we calculated subtype abundance for each batch. We then used the filterByExpr function to filter out subtypes with low-abundance. The negative binomial dispersion was then estimated for each subtype with the estimateDisp function with trend = none. Then the quasi-likelihood dispersion was computed using glmQLFit with abundance. trend = false. We used glmQLFTest to perform empirical Bayes quasi-likelihood F-tests to test for significant differences in subtype abundance between the different conditions. We show just the differences between Uninjured Day 1 and each other dataset. We then calculated differential abundance using Milo, which assesses abundance on K-nearest neighbor graph [37]. We converted our Seurat object into a Single Cell Experiment. We then built a Milo graph for each comparison, UI/I1, U1/I7, and U1/U7. We calculated relative neighborhood occupancy, plotted this KNN analysis, as well as beeswarm plots of subtype associated neighborhood changes.

## GO enrichment analysis

GO enrichment analysis was performed on the top differentially expressed genes in each dataset as defined by a log fold change of greater than 1 with a P value of < 0.001. Gene lists were then submitted to ShinyGO 0.80 using the Zebrafish library and considered relevant with a false discovery rate below 0.01 [131].

## Pseudotime analysis

We performed trajectory analysis using Monocle3 version 1.3.7. We ordered cells within pseudotime using the subtype 3 cluster as the root for the analysis. This analysis was performed according to the Monocle3 trajectory vignette. https://cole-trapnell-lab.github.io/monocle3/docs/trajectories/.

## Assessment of differentiation state

We converted gene names from fish to mouse using BiomaRt 2.60.1 referencing the February 2021 ensemble archive https://feb2021.archive.ensembl.org [132,133]. We then performed cell potency analysis using CytoTRACE 2 version 1.0.0 as described in the CytoTRACE2 vignette [48].

## Bulk RNA-seq analysis

5 retinas per sample (injured and uninjured at 1, 5 and 7 dpi) were dissected and dissociated per experiment and repeated independently three times for biological replicates. One thousand GFP + RGCs were collected in the SmartSeq HT kit (Takara Bio) lysis buffer. Library preparation, quality control analysis, and next-generation sequencing were performed by the Health Sciences Sequencing Core at Children's Hospital of Pittsburgh as previously described [134]. cDNA sequencing libraries were prepared using a SmartSeq HT kit (Takara Bio) and Illumina Nextera XT kit (Illumina Inc.) and 2 × 100 paired end, 200 cycle sequencing was performed on a NextSeq 2000 system (Illumina), aiming for 40 million reads per sample. After sequencing, raw read data were imported to the CLC Genomics Workbench (Qiagen Digital Insights) licensed through the Molecular Biology Information Service of the Health Sciences Library System at the University of Pittsburgh. After mapping trimmed reads to the *Danio rerio* reference genome (assembly GRCz11), DEGs from 1, 5 and 7 dpi time points were identified using the following filter: the maximum of the average group RPKM value >1.5, absolute fold change >1.5, FDR P-value <0.05. Genes with TPM = 0 in one or more replicates were excluded.

## Tissue processing

Fish were euthanized, decapitated and fish heads fixed in 4% paraformaldehyde overnight after puncturing the cornea with a 30G needle. Fixed tissue was processed in ice cold PBS after pinning the fish head onto a silicon coated petri plate. Surgical microscissors were used to make a radial cut across the cornea followed by cuts around the circumference of the cornea. The lens and vitreous fluid were gently dislodged from the eye cup using forceps. Two forceps were used to slightly rip the sclera apart at one edge and loosen the retina. The eye cup was gently pulled out of the socket by pulling on the sclera or the optic nerve. The retina was then peeled from the sclera and the choroid around it. Once the retina was sufficiently cleaned, it was transferred from PBS to PBST (PBS + 0.1% tween) in a 24 well on ice using a 1mL pipette tip cut at the end. The harvested tissue was washed three times in PBST on a nutator at 4°C followed by two 100% methanol (MeOH) washes before being left in 100% MeOH overnight at -20°C.

## TUNEL staining

Cell death was accessed using the Biotium CFR640R TUNEL Assay Apoptosis Detection Kit. Whole retinae were extracted from ONT fish at day 1 and day 7 post injury and NMDA-injected and control fish at day 1 post injury and processed as described above. Retinae were rehydrated with serial dilutions of methanol (MeOH) and Phosphate Buffered Saline with 0.1% Triton-X(PBTx) on ice. The tissue was then treated with 25g/mL solution of proteinase K for 5 mins and fixed with 4% PFA for 20 mins. After washing with PBTx, retinae were incubated with the TUNEL solution (10µL of enzyme in 90µL buffer) for 3h at 37°C with gentle agitation. Retinae were washed with PBTx and post fixed with 4% PFA for 30 mins and then washed with PBTx before being flat mounted on slides for imaging.

## *In situ* hybridization

*In situ* hybridization chain reaction (HCR) v3.0 with split initiator probes (Molecular Instruments) was performed following [135]. Briefly, the processed tissue was rehydrated in a MeOH gradient, treated with proteinase K (2g/mL) and refixed in 4% paraformaldehyde. Each probe (2µL) was added to 500µL of hybridization buffer per well containing 3–5 retinae and incubated overnight at 37°C. The following day, the retinae were washed in the wash buffer, PBST and then SSCT (saline sodium citrate + 0.1% tween). Snap cooled amplifiers (10L of hairpin1 and hairpin2) in the amplification buffer were added to the samples and incubated overnight in a dark box at room temperature. Before flat mounting, retinae were washed thoroughly in SSCT and stained with DAPI (1:250).

## Confocal microscopy, image processing and quantification

Retinal flat-mounts were prepared as above and images were taken using an upright Nikon AXR laser scanning microscope. The whole retina large image was acquired at 20X using ND acquisition mode which allows a defined starting point (XYZ), number of fields to be covered in the X and Y direction, and a symmetric Z series relative to the starting point (Z). Laser and gain settings were optimized for the injured retina at day 7 and kept constant across all experimental conditions. The captured fields were stitched by the Nikon AXR NIS software using default settings (15% blended). The acquired images were processed using the Extended Depth of Focus (EDF) feature of the Nikon AXR NIS software to obtain a 2D image. For quantification of HCR-FISH signal, images were captured at a higher resolution using a 60X objective and 1.7X zoom. Images were taken from each of the 4 quadrants (1 peripheral, 1 central per quadrant) and Z series across the RGC layer (2.5µM). Quantification was performed using Imaris 9.6.0 (Bitplane). Raw images (.nd2) were first converted into Imaris files and then analyzed using the Spot Function on machine learning mode. Measurements were exported and statistically assessed using Prism 10.0 (Graphpad) using non-parametric one-way analysis and Kruskul-Wallis post hoc testing. For quantification of cells in the ganglion cell layer, DAPI images were captured using a 40X objective and 1X zoom. Two images were taken from each quadrant (1 peripheral image and 1 central image per

quadrant), yielding 8 images per retina. Images were processed using FIJI ImageJ to obtain 2D images and quantified using the Trainable Weka segmentation plugin.

## Supporting information

**S1 Table. Differentially expressed genes across RGC subtypes.** Statistical analysis determining differentially expressed genes found within each RGC subtype.
(XLSX)

**S2 Table. Differential abundance statistical analysis of RGC subtypes across experimental conditions.** Statistical analysis determining differential abundance of subtype representation between Uninjured Day 1 and Injured Day 1, Uninjured Day 1 and Injured Day 7, Uninjured Day 1 and Uninjured Day 7.
(XLSX)

**S3 Table. Differentially expressed genes across subclustered subtype 3 RGCs.** Statistical analysis determining differentially expressed genes found within each subclustered subtype 3 RGC, including GO and KEGG analysis, if found statistically significant.
(XLSX)

**S4 Table. Differentially expressed genes across pseudobulk analysis of each experimental condition.** Statistical analysis for differentially expressed genes enriched in each experimental condition, Uninjured Day 1, Uninjured Day 7, Injured Day 1 and Injured Day 7.
(XLSX)

**S1 Fig. Analysis of data integrated with the Kölsch etal., 2021 dataset.** We performed a parallel analysis to assess our data relative to a previously published adult zebrafish wildtype RGC single cell dataset [25]. We found that an integrated cluster analysis showed representation of each of the five datasets (Wildtype, Uninjured Day 1, Uninjured Day 7, Injured Day 1 and Injured Day 7) in all clusters. Although we detect an uninjured eye response in our Uninjured Day 7 dataset (Fig 7), we found that our Uninjured Day 1 dataset showed consistent similarities to the wildtype, uninjured dataset [25]. We therefore chose to use our Uninjured Day 1 dataset as our control comparison. By only using data generated in the current study, we avoided some batch effects in our experiment relative to the previously published data. This figure shows the similarities between previously published wildtype data and the Uninjured Day 1 data. A) Uniform manifold approximation and projection (UMAP) of all data generated in this study integrated with the Kölsch et al., 2021 dataset [25]. This independent analysis identified 32 clusters [25]. B) Differential abundance bar graph of each subtype across the datasets, including the Kölsch et al., 2021 dataset labeled as wildtype. C) The integrated UMAP of data generated in this study and previously published wildtype adult data, split by each dataset: Wildtype [25], Uninjured Day 1, Uninjured Day 7, Injured Day 1 and Injured Day 7. D,E) Milo plots for differential abundance comparison. Milo analysis shows differential occupancy of cells in KNN graphs. This is a Beehive plot showing the differential abundance of each subtype. Each dot is a neighborhood and the dots toward the right (red) are neighborhoods enriched in the Injured Day 7 dataset. The dots toward the left are enriched in the Wildtype or Uninjured Day 1 dataset respectively. The assigned cluster number is on the left. Mixed, means the neighborhood occupies two clusters D) Comparison of wildtype versus Injured Day 7 and E) Comparison of Uninjured Day 1 versus Injured Day 7. F,G) GO enrichment analysis using the top 1500 differentially expressed genes between Wildtype/Injured Day 7 and Uninjured Day 1/Injured Day 7 using the Wilcoxon rank sum test. These are all the KEGG terms with an FDR of < 0.01 and a Fold Enrichment of > 1.5. Analysis was performed using SingyGO 0.82 using the Zebrafish library. The KEGG terms shared between the two analyses are colored in blue.
(TIF)

**S2 Fig.  Quality control for single cell analysis.** A) Percent mitochondrial genes per cell separated by replicate and organized by sample showing low percentage of mitochondrial genes overall. B) UMAP representation of each replicate, organized by experimental condition showing consistency across replicates C) UMAP representation of each experimental condition grouped by replicate. This representation shows that each cluster includes cells from each replicate. D) UMAP representation of all merged uninjured data, both Uninjured Day 1 and Uninjured Day 7. 31 clusters were found. The colors do not correlate with our final integrated analysis shown in Fig 1. E) UMAP representation of all injured data, both Injured Day 1 and Injured Day 7. 24 clusters were found. The colors do not correlate with the final integrated analysis shown in Fig 1. E and 1F are a result of a merged, not integrated, pipeline. F) Violin plot of *isl2b* expression in pseudobulk analysis of Uninjured Day 1, Uninjured Day 7, Injured Day 1 and Injured Day 7 RGCs showing downregulation after injury. (TIF)

**S3 Fig.  Feature plots for RGC subtype markers.** Corresponding feature plots with a min.cuttoff of q25 for the RGC subtype markers shown in Fig 1D. (TIF)

**S4 Fig.  RGCs are not apoptotic at 1 day post ONT.** Representative images of high resolution fields of view at 1 day post ONT, naive, and NMDA injured retina (40X objective, n = 24 fields of view per condition). A) Injured retina at 1 dpi do not have TUNEL$^+$ cells. B) Uninjured and C) naive retina also show no TUNEL$^+$ cells. D) NMDA injury elicits substantial TUNEL$^+$ RGCs at 1 dpi compared to E) PBS controls. (TIF)

**S5 Fig.  RGCs are not apoptotic at 7 days post ONT.** Representative images of high resolution fields of view at 7 day post ONT, naive, and NMDA injured retina (40X objective, n = 24 fields of view per condition). A) Injured retina at 7 dpi do not have TUNEL$^+$ cells. B) Uninjured and C) naive retina also show no TUNEL$^+$ cells. D) NMDA injury elicits substantial TUNEL$^+$ RGCs at 7 dpi compared to E) PBS controls. (TIF)

**S6 Fig.  Quantification of RGC nuclear number and size in response to ONT.** Representative images of high resolution fields of view in 7 days post ONT or naive retinae (40X objective, n = 32 fields of view per condition) show that A) RGC density decreases 7 days post injury compared to naive. B) Injured retinae at 7 dpi have fewer nuclei (M = 1958, SD = 385) compared to uninjured (M = 2304, SD = 616) and naive (M = 2460, SD = 651) retinae. C) Nuclei of injured RGCs at 7 dpi are larger (M = 33, SD = 3) than those in uninjured (M = 26, SD = 3) or naive (M = 24, SD = 2.7) retina. DAPI stains mark nuclei. Scale bar = 50µm. (TIF)

**S7 Fig.  Enlarged feature plot showing increased detail with a min.cuttoff of q25 corresponding to Fig 4A.** (TIF)

**S8 Fig.  Enlarged feature plot showing increased detail with a min.cuttoff of q25 corresponding to Fig 8D.** (TIF)

**S9 Fig.  Expression of *tubb5* and *alcamb* is robust by 3 days post ONT.** Representative images of high resolution fields of view in 3 and 5 days post ONT or naive retinae (40X objective, n = 16–24 fields of view per condition. n = 3 retina per condition) show that A) *tubb5* and *alcamb* expression in RGCs is robust throughout the retina at both 3 and 5 days post injury B) *tubb5* expression is significant as early as 3 dpi compared to uninjured and naive controls and increases significantly at 5 dpi. C) Expression of *alcamb* is significant as early as 3 dpi compared to uninjured and naive controls and increases significantly at 5 dpi. (TIF)

**S10 Fig. Expression of *tubb5* and *alcamb* is transient in response to ONT.** Representative images of high resolution fields of view at 7, 25 and 50 days post ONT or naive retina (60X objective, n = 24 fields of view per condition) show that A) Injured retina at 7 dpi show robust and uniform expression of *tubb5* and *alcamb*. B) Uninjured retina at 7 dpi express *tubb*5 and *alcamb*, albeit dispersed. C-F) Expression of *tubb5* and *alcamb* return to baseline levels at 25 and 50 dpi, similar to G) baseline expression in naive retina.
(TIF)

## Acknowledgments

We thank Dr. Tony St. Leger (The University of Pittsburgh Medical School) for his expertise and assistance with flow assisted cell sorting, Shelby Crowley for help with HCR-FISH and Beth Gibbons for editorial assistance.

## Author contributions

**Conceptualization:** Takaaki Kuwajima, Kristen M. Koenig, Jeffrey M Gross.

**Data curation:** Hannah Schriever, Kristen M. Koenig.

**Formal analysis:** Ashrifa Ali, Hannah Schriever, Takaaki Kuwajima, Kristen M. Koenig, Jeffrey M Gross.

**Funding acquisition:** Ashrifa Ali, Jeffrey M Gross.

**Investigation:** Ashrifa Ali.

**Project administration:** Jeffrey M Gross.

**Supervision:** Dennis Kostka, Jeffrey M Gross.

**Validation:** Ashrifa Ali, Kristen M. Koenig.

**Visualization:** Kristen M. Koenig.

**Writing – original draft:** Ashrifa Ali, Kristen M. Koenig, Jeffrey M Gross.

**Writing – review & editing:** Ashrifa Ali, Hannah Schriever, Dennis Kostka, Takaaki Kuwajima, Kristen M. Koenig, Jeffrey M Gross.

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
