## [Decision Letter · Decision Letter 0]

25 May 2025

PGENETICS-D-25-00397

Zebrafish optic nerve injury results in systemic retinal ganglion cell dedifferentiation

PLOS Genetics

Dear Dr. Gross,

Thank you for submitting your manuscript to PLOS Genetics. After careful consideration, we feel that it has merit but does not fully meet PLOS Genetics's publication criteria as it currently stands. Therefore, we invite you to submit a revised version of the manuscript that addresses the points raised during the review process.

Please submit your revised manuscript within 60 days Jul 24 2025 11:59PM. If you will need more time than this to complete your revisions, please reply to this message or contact the journal office at plosgenetics@plos.org. Please include the following items when submitting your revised manuscript:

We look forward to receiving your revised manuscript.

Kind regards,

Brian D. Perkins, Ph.D.

Academic Editor

PLOS Genetics

Fengwei Yu

Section Editor

PLOS Genetics

Aimée Dudley

Editor-in-Chief

PLOS Genetics

Anne Goriely

Editor-in-Chief

PLOS Genetics

**Additional Editor Comments :**

Dear Jeff,

First, apologies for the delays in returning the manuscript reviews. As you can see, all three reviewers feel that the manuscript presents interesting findings but that additional work is needed. Importantly, the reviewers feel the manuscript in its current form is largely descriptive and additional mechanistic insights are necessary to fully support the findings.

**Journal Requirements:**

At this stage, the following Authors/Authors require contributions: Ashrifa Ali, Hannah Schriever, Dennis Kostka, Takaaki Kuwajima, Kristen Koenig, and Jeffrey M Gross. Please ensure that the full contributions of each author are acknowledged in the "Add/Edit/Remove Authors" section of our submission form.

The list of CRediT author contributions may be found here: https://journals.plos.org/plosgenetics/s/authorship#loc-author-contributions

4) We notice that your supplementary Figures are included in the manuscript file. Please remove them and upload them with the file type 'Supporting Information'. Please ensure that each Supporting Information file has a legend listed in the manuscript after the references list.

Potential Copyright Issues:

i) Figure 1A. Please confirm whether you drew the images / clip-art within the figure panels by hand. If you did not draw the images, please provide (a) a link to the source of the images or icons and their license / terms of use; or (b) written permission from the copyright holder to publish the images or icons under our CC BY 4.0 license. Alternatively, you may replace the images with open source alternatives. See these open source resources you may use to replace images / clip-art:

6) Thank you for stating "All sequencing data have been deposited to the NCBI Gene Expression Omnibus (GEO) under the following accession numbers: RNA-Seq:GSE284728 scRNA-Seq:GSE284729." Please note that, though access restrictions are acceptable now, your entire minimal dataset will need to be made freely accessible if your manuscript is accepted for publication. This policy applies to all data except where public deposition would breach compliance with the protocol approved by your research ethics board. If you are unable to adhere to our open data policy, please kindly revise your statement to explain your reasoning and we will seek the editor's input on an exemption.

7) Please ensure that the funders and grant numbers match between the Financial Disclosure field and the Funding Information tab in your submission form. Note that the funders must be provided in the same order in both places as well. Currently, "the University of Texas at Austin" is missing from the Funding Information tab.

**Reviewers' comments:**

Reviewer's Responses to Questions

Reviewer #1: The paper by Ali and colleagues characterizes patterns of gene expression in injured retinal ganglion cells (RGCs) in the zebrafish retina. Different scRNA-seq were generated for FACS-enriched isl2:GFP-positive RGCs from control and ON-injured retinas. UMAP-ordering and label transfer for established RGC subtypes revealed treatment-dependent differential distribution of among RGC subtypes and k- means neighborhoods. RGC subtype 3 was identified as the least mature cluster of RGCs. Pseudotime and cytoTRACE trajectories originating from cluster 3 were established to identify the distribution of differentiating/maturing RGCs across the UMAP dimensional reduction. Figures 8 and 9 include FISH validation for tubb5 and alcamb in normal and injured retinas with isl2b:GFP RGCs in flat-mounts.

The paper is entirely descriptive, but contains several interesting observations.

Comments for the authors to consider:

- Was there any compelling rational to apply 40 principle components (PCs; dims=40)? Was a JackStraw or Elbow plot used to identify a reasonable cutoff for the number of PC’s with high variability?

- Was there any compelling rational to use IntegratedAnchor and IntegrateData functions to robustly normalize the different libraries? Why not normalize ("aggr" tool) in Cell Ranger or merely used the default Seurat normalization since the libraries were generated, presumbly, with the one Cell Controller run and and identical reagents.

- Was there any filtering for high mitochondrial content to remove dying cells? If so, what was the cut-off?

- Since FindVariableFeature was run with a cut-off at only nfeatures = 1500, it seems unlikely that 40 PCs with a high variability were obtained to justify the approach, unless the average nfeatures per cell was very high. Or, was this lower cuttoff chosen because V2 reagents didn't capture high features/cell?

- There are no statistics provided for lists of differentially expressed genes (DEGs). It seems necessary to include supplemental data as spreadsheets for the findmarkers output with adjusted p values and logFC for the lists of DEGs for the different comparisons.

Figure 1: The table/legend with the cluster number and color is not useful and could be removed. It would be helpful to include an occupancy histogram illustrating the percent occupancy for each orig.ident in each UMAP cluster.

Figure 1: For consistency, it would be helpful to maintain the same color scheme for the dotplots in panel E as in panel D.

Figure 2: I have no idea what the lines connecting the different k-neighbors in the Milo analysis represent.

Figure 3: The feature plots in 3A are difficult to see even with digital zoom. Further these plots might benefit from a min.cutoff and max.cutoff to clean-up and better highlight the highly expressing cells. This comment applies to all other feature plots.

Figure 3: Panel E makes my head hurt – clearly RGC subtype 3 has a distinct transcriptomic profile. The list of genes along the X axis is too expansive and all of the rows of low-expressing RGC subtypes are dead space. These plot could be simplified – perhaps create a new column of metadata for cluster3 vs others to collectively highlight the average expression and percent expression of representative favorite genes. Please choose some of your favorites genes to populate the X axis that perhaps have been previously implicated in important developmental processes. I acknowledge that scRNA-seq provides a deep wealth of data and it is challenging to pick the most relevant and meaningful and favorite genes. It might also help to group and indicate some of the representative favorite genes according to GO category. Again, supplemental data with findmarkers stats would help to further convince readers of the distinct patterns of gene expression. This comment applies to all dotplots.

Figure 4: It would be helpful to include headings for each plot and labels for the X and Y axes in panels B and C. It is not possible to interpret these data without carefully reading the legends and methods, which creates more work for lazy readers (like me).

Figure 4: the cytoTRACE and pseudotime trajectories are not consistent with respect to clusters 7, 8 and 23. The pseudotime appears to place these clusters earlier in pseudotime, whereas the cytoTRACE places these clusters as more differentiated. It would be helpful to impart some biological commons sense to and ask whether a handful of genes that are well-established to be highly expressed by happy mature RGCs correlate better with pseudotime trajectory endpoints or cytoTRACE more differentiated scores.

Figure 6 and 7: It is not clear that the pseudo-bulk analysis is very different than generating lists of DEGs by findmarkers by orig.ident. If something distinctly different is revealed by pseudo-bulk (which may provide more statistical power), then this should be better highlighted.

Figure 7: panel C: please include stats for these genes in the violin plots by orig.ident. A supplemental table would be acceptable.

Discussion: “specific loci in the contralateral uninjured eye show similar transcriptional responses to that in the injured eye?” I am not sure that “specific loci” is the appropriate term. Perhaps “sets of genes” is better. There was no gene network analysis or chromatin domain interactions inferred from the RNA-seq data.

Reviewer #2: In this manuscript, Ali et al. use zebrafish as a model to study retinal ganglion cell (RGC) resilience and regeneration, an important scientific direction. In mammals, including humans, RGCs degenerate and cannot regenerate, leading to irreversible blindness in diseases like glaucoma. Zebrafish RGCs are known to be both resilient and regenerative after optic nerve injury, so they are an appropriate model for this study. In mice, scRNA-seq has shown important heterogeneity between RGC subtypes after optic nerve injury, the authors address whether the same is true in zebrafish. Using a combination of RNA-sequencing and bioinformatic methods, the authors identify a putatively regenerative, immature RGC type (Type 3) and show evidence that following injury, there is broad dedifferentiation of RGCs to become similar to Type 3 RGCs. While intriguing, the bioinformatic results are not fully supported by robust in vivo data. Confidence in the in silico data is also hampered by a lack of description of how the methods were performed. These concerns and others are enumerated below.

Figure 1: Major Point: The UMAP shown in Figure 1B drives much of the later focus on cluster 3. Could the authors please expand on how these samples were merged/integrated and if the subtype assignments are the same as those shown in Supp Figure 1? The authors mention that they started with >100,000 cells but end up with only ~17,000 RGCs. Were these other ~80,000 cells non-RGCs or were they removed because they were low quality? Please clarify these details about the UMAP and add additional detail about quality control and the excluded cells. Separate UMAPs (non-merged/non-integrated) from injured and uninjured RGCs would also be helpful.

Minor points: Could the authors please add language justifying the use of their time points for collecting RGCs post-injury? It would be helpful if they were contextualized for unfamiliar readers.

The sentence “it has been shown that specific subtypes are susceptible…” is a bit misleading because it suggests that some mouse RGC types do not die after injury. In Tran et al. it is shown that all RGC types die, but the proportion of cells that die in a given type varies. If the authors agree, they should clarify this point.

The authors should contextualize the RGC types as much as possible. The cluster numbers are difficult to interpret, so if there is anything known, or that can be gleaned from their data, that would help compare with RGC types in other species, that would be helpful.

I cannot find details in the manuscript about when the NMDA injected fish were collected. Please add this detail.

Figure 2: Major Point: The increase to cluster 3 abundance is the central point of the paper. Could you please elaborate on how many replicates were collected, whether they were biological or technical, and how consistent the replicates were? It is a little bit surprising the 1 dpi retinas already have a much higher frequency of C3 RGCs than the uninjured 1 dpi retinas.

It is intriguing that isl2b is downregulated and that there is no TUNEL staining detected after optic nerve injury, but would this method be sufficient to detect the ~20% loss of RGCs previously reported? If the authors could show no change to RGC number with an alternative marker this would be more convincing, or they could remove the claim of no RGC loss.

Figure 3: Major Point: Is this DE analysis being conducted between C3 RGCs and other RGCs across all conditions? If so would these results contain a mix of C3 baseline properties and injury response programs? Please clarify and consider doing separate DE analysis for injured and uninjured C3 RGCs.

Minor point: Please add a citation for the references to the regeneration and immature markers genes tubb5, alcamb, tmsb, prph, sox11a…

Figure 5: Major Point: Subclustering of C3 seems to reveal four subclusters that express markers of other subtypes. This suggests that the initial C3 assignment may not be accurate. If these four subclusters are removed, is there still a significant increase in C3 abundance? Can the authors please show individual expression profiles for these clusters to demonstrate more clearly to what extent they express C3 marker genes vs. other subtype marker genes.

The change in Type 3 RGCs in the uninjured day 7 eyes is intriguing. The principal issue with any “contralateral effect” is how the uninjured eye senses the injury to the fellow eye. Previous work in the mouse showed that networked astrocytes redistribute metabolic resources during stress (Cooper et al. 2020, PNAS DOI: 10.1073/pnas.2009425117). The authors should add data showing whether contralateral glia respond in their model or propose another mechanism.

Without evidence of a mechanism, there is possibility that these changes are a batch effect of sample collection and processing. Were the seven-day samples collected and sequenced separately from the day one samples? The authors should add clarifying language to the methods and results section.

Figure 6: Minor points: The authors discuss GO terms which is interesting, but it’s unclear what the novelty of these findings are and how they relate to models where regeneration does not occur. They authors should change this part to highlight what is novel about their findings and how they are unique to the regenerating retina.

Figure 7: Minor points: This GO and KEGG terms that pop out of their analysis lack novelty and don’t add to the story of the paper. This figure could be moved to supplementary materials.

Figure 8: Major point: The variability of signal of the different markers across the retina is concerning. Is there a reason that the staining would look so patchy? The explanation in the text is lacking. The authors say there are “concentric circles of RGCs emanating form the optic disc” but the images provided appear to show a stochastic pattern of staining that is difficult to interpret.

It’s not clear what the arrows are pointing to because the entire retinal whole mount is shown. Because this figure presents some of the only biological affirmation of the bioinformatic results, it is important to support the authors’ overall conclusions.

Figure 9: Minor points: It is nice that the authors show uninjured day 7 regions that show variable expression of tubb5 and alcamb. The authors should make a more direct reference to this in the body of the text. In its current form, the manuscript contains very little description of Figure 9 or its relevance.

Reviewer recommendation: Overall, this manuscript provides intriguing data about how zebrafish RGCs survive and assume a more regenerative identity after injury. The manuscript is well written, but some details need to be added and the relevance of some of the bioinformatic results is not clear. Furthermore, the histologic results are shaky and must be improved in order to support the intriguing in silico data. Therefore, I recommend revisions prior to publication.

Reviewer #3: The manuscript by Ali et al. characterized the retinal ganglion cell (RGC) injury response following optic nerve transection in adult zebrafish retinas. Using a combination of single-cell RNA sequencing (scRNA-seq) and histological analyses, the authors showed that all RGC types survive injury and adopt a dedifferentiated, immature-like state. Interestingly, the study also shows that RGCs in the contralateral, uninjured eye also exhibit an injury response, albeit limited to specific regions and at a lower level. This work offers new insights into potential molecular targets for promoting RGC survival in optic neuropathies. However, the manuscript remains largely descriptive and would be significantly strengthened by addressing the following points:

1. The authors reported an increase in the proportion of RGC type 3 following injury and concluded that all zebrafish RGCs survive damage by temporarily shifting into a less mature state, resembling a rare population of immature RGCs found in uninjured animals. This conclusion is not fully supported with current data, given that: 1) Only two timepoints were assessed by scRNA-seq and RNA-FISH; 2) Most other RGC clusters remain clearly distinct from type 3; 3) The conclusion is based primarily on the expression of a few marker genes (e.g., tubb5, alcamb, tmsb5).

An alternative explanation is that injured RGCs share expression of a limited set of genes with type 3, rather than fully transitioning into this state.

2. To what extent do RGCs adopt the type 3 state after injury? Do all RGC types undergo this transition, or only a subset? Analysis of additional intermediate timepoints would clarify the dynamics of this potential state transition.

3. It remains unclear whether adoption of the type 3 state actively promotes RGC survival or is merely a by-product of injury. What factors drive RGCs to adopt the type 3 state? Functional testing of candidate regulators would significantly strengthen the manuscript's mechanistic depth.

4. Where are type 3 RGCs located in wild-type retinas? Could they represent immature RGCs continuously generated from the ciliary marginal zone (CMZ)?

5. RNA-FISH analysis shows an increase in type 3 markers (tubb5, alcamb) from day 1 (low expression) to day 7 (high and widespread expression) post-injury. Including an intermediate timepoint would improve understanding of expression dynamics.

6. In several plots and heatmaps, the label font size (e.g., cluster numbers, gene names) is too small, making the data difficult to interpret. Increasing the font size would improve readability.

**Have all data underlying the figures and results presented in the manuscript been provided?**

Reviewer #1: Yes

Reviewer #2: Yes

Reviewer #3: Yes

PLOS authors have the option to publish the peer review history of their article (what does this mean? ). If published, this will include your full peer review and any attached files.

**Do you want your identity to be public for this peer review?** For information about this choice, including consent withdrawal, please see our Privacy Policy .

Reviewer #1: No

Reviewer #2: No

Reviewer #3: No

**Figure resubmission:**
---

## [Decision Letter · Decision Letter 1]

11 Sep 2025

Dear Dr Gross,

We are pleased to inform you that your manuscript entitled "Zebrafish optic nerve injury results in systemic retinal ganglion cell dedifferentiation" has been editorially accepted for publication in PLOS Genetics. Congratulations!

Yours sincerely,

Brian D. Perkins, Ph.D.

Academic Editor

PLOS Genetics

Fengwei Yu

Section Editor

PLOS Genetics

Aimée Dudley

Editor-in-Chief

PLOS Genetics

Anne Goriely

Editor-in-Chief

PLOS Genetics

Comments from the reviewers (if applicable):

Reviewer #1:

Reviewer #2:

Reviewer #3:

Reviewer's Responses to Questions

**Comments to the Authors:**

Reviewer #1: The revisions are satisfactory. The reviewers have adequately addressed the comments from the reviewers.

Reviewer #2: In this revised manuscript, the authors have addressed all of my concerns.

Reviewer #3: I appreciate that the authors' effort to address three reviewer's comments and revise the manuscript. The manuscript is significantly improved. Congrats to the authors

**Have all data underlying the figures and results presented in the manuscript been provided?**

Reviewer #1: Yes

Reviewer #2: None

Reviewer #3: Yes

PLOS authors have the option to publish the peer review history of their article (what does this mean? ). If published, this will include your full peer review and any attached files.

**Do you want your identity to be public for this peer review?** For information about this choice, including consent withdrawal, please see our Privacy Policy .

Reviewer #1: No

Reviewer #2: No

Reviewer #3: No

**Data Deposition**

http://datadryad.org/submit?journalID=pgenetics&manu=PGENETICS-D-25-00397R1

**Press Queries**

---

## [Editor Report · Acceptance letter]

PGENETICS-D-25-00397R1

Zebrafish optic nerve injury results in systemic retinal ganglion cell dedifferentiation

Dear Dr Gross,

We are pleased to inform you that your manuscript entitled "Zebrafish optic nerve injury results in systemic retinal ganglion cell dedifferentiation" has been formally accepted for publication in PLOS Genetics! Your manuscript is now with our production department and you will be notified of the publication date in due course.

With kind regards,

Anita Estes

PLOS Genetics

On behalf of:
